# Connexin 41.8 governs timely haematopoietic stem and progenitor cell specification

Tim Petzold[1,2,3], Sarah Brivio[1], Tanja Linnerz[1,4], Masakatsu Watanabe[5], Holger Gerhardt[2,3,6] and Julien Y. Bertrand[1,7,*]

## ABSTRACT

Haematopoietic stem and progenitor cells (HSPCs) derive from a subset of endothelial cells (ECs), known as haemogenic ECs by the process of endothelial-to-haematopoietic transition (EHT). Although many factors involved in EHT have been elucidated, we still have a poor understanding of the temporal regulation of this process. Mitochondrial-derived reactive oxygen species (ROS) have been shown to stabilise the hypoxia-inducible factor $1/2\alpha$ (Hif1/2$\alpha$) proteins, allowing them to positively regulate EHT. Here, we show a developmental delay in EHT and HSPC induction in a *connexin (cx)41.8* (orthologous to mammalian *CX40*) gap junction mutant, in zebrafish. In mammalian cells, CX40 has been shown to localise to the mitochondria. We demonstrate that Cx41.8 is important for the correct temporal generation of mitochondrial ROS, which stabilise the Hif pathway, allowing for the subsequent specification of the haemogenic endothelium. Taken together, our data indicate that Cx41.8 governs the correct temporal induction of HSPCs.

KEY WORDS: Cx41.8, CX40, HSPC, ROS, Zebrafish

## INTRODUCTION

Haematopoietic stem and progenitor cells (HSPCs) are rare, highly specialised cells that sit at the top of the haematopoietic hierarchy. HSPCs have the ability to self-renew and give rise to progenitor cells, which differentiate into mature blood cells (Laurenti and Gottgens, 2018). In vertebrates, HSPCs derive from the haemogenic endothelium (Ottersbach, 2019), in a highly conserved process known as endothelial-to-haematopoietic transition (EHT) (Bertrand et al., 2010; Boisset et al., 2010; Kissa and Herbomel, 2010; Eilken et al., 2009). Having a detailed understanding of all the factors involved in EHT may allow for *in vitro* generation of HSPCs from

endothelial cells (ECs) in the future, which could have significant implications for regenerative medicine.

Connexin proteins play diverse roles in health and disease (Laird and Lampe, 2018). Six connexins form a connexon, which, when present at the cell membrane, can dock onto a connexon on a neighbouring cell to form a gap junction (Goodenough and Paul, 2009). Gap junctions can facilitate the transport of ions, amino acids and small metabolites across the plasma membrane (Delmar et al., 2018). Interestingly, in mammals, CX40, along with other connexins, has been found to localise to the membrane of intracellular organelles such as the mitochondria (Guo et al., 2017; Boengler et al., 2022), where it promotes the production of reactive oxygen species (ROS) (Guo et al., 2017). We previously demonstrated that zebrafish *cx41.8* (orthologous to mammalian *CX40*) plays a role in HSPC expansion in the caudal haematopoietic tissue (Cacialli et al., 2021). Indeed, HSPCs died by apoptosis in *cx41.8^{t1/t1}* mutants as a result of ROS toxicity during their expansion phase, whereas their specification and emergence were unaffected (Cacialli et al., 2021).

Here, we find that another *cx41.8* zebrafish mutant, *cx41.8^{tq/tq}*, has a delay in the specification of the haemogenic endothelium resulting in delayed HSPC emergence. We determine that this phenotype is mechanistically linked with mitochondrial ROS production and the hypoxia-inducible factor (Hif) pathway. We suggest that mitochondrial Cx41.8 contributes to the correct temporal induction of EHT and the subsequent formation of HSPCs during zebrafish development.

## RESULTS

### *cx41.8^{tq/tq}* mutants harbour an HSPC specification defect

We previously characterised definitive haematopoiesis in the *cx41.8^{t1/t1}* mutant (Cacialli et al., 2021) and decided to investigate whether the *cx41.8^{tq/tq}* mutant displays the same haematopoietic phenotype. The *leo^{tq270}* mutant (referred to as *cx41.8^{tq/tq}* throughout), possesses a missense mutation, I203F, in the fourth transmembrane domain of the protein (Watanabe and Kondo, 2012) (Fig. 1A,B), which results in disruption of the channel function (Watanabe et al., 2006, 2016). Primitive haematopoiesis was found to be unaffected in *cx41.8^{tq/tq}* embryos relative to controls, as determined by the expression of *gata1* and *pu.1* (primitive erythrocytes and primitive myeloid cells, respectively), by whole-mount *in situ* hybridisation (WISH) at 24 hpf (Fig. S1A,B). However, markers of definitive haematopoiesis were found to be significantly altered in *cx41.8^{tq/tq}* embryos: *gata2b* expression at 24 hpf (Fig. 1C), and *runx1* expression at 24 hpf (Fig. 1D) and 28 hpf (Fig. S2A) were reduced when compared with controls, as determined by WISH. In addition, a reduction in the number of *cmyb^+* haemogenic endothelial cells on the ventral side of the dorsal aorta in *cmyb:GFP* embryos was observed in *cx41.8^{tq/tq};cmyb:GFP* embryos (see Fig. S3A for representative image) at 28 and 30 hpf (Fig. 1E), providing further evidence for a defect in HSPC specification in *cx41.8^{tq/tq}* embryos.

[1]University of Geneva, Faculty of Medicine, Department of Pathology and Immunology, Rue Michel-Servet 1, Geneva 4, Switzerland. [2]Max Delbrück Center for Molecular Medicine in the Helmholtz Association, Berlin, Germany. [3]DZHK (German Center for Cardiovascular Research), Partner Site Berlin, Berlin, Germany. [4]University of Auckland, Faculty of Medical and Health Sciences, Department of Molecular Medicine and Pathology, 85 Park Road, 1023 Auckland, New Zealand. [5]Cellular and Structural Physiology Institute (CeSPI), Nagoya University, Furo-cho, Chikusa-ku, Nagoya, 464-8601, Japan. [6]Charité - Universitätsmedizin Berlin, Berlin, Germany. [7]Geneva Centre of Inflammation Research, University of Geneva, Faculty of Medicine, Rue Michel-Servet 1, Geneva 4, Switzerland.

*Author for correspondence ( julien.bertrand@unige.ch)

T.P., 0000-0001-9788-6167; S.B., 0000-0003-1913-8686; T.L., 0000-0001-8127-780X; M.W., 0000-0002-5744-197X; H.G., 0000-0002-3030-0384

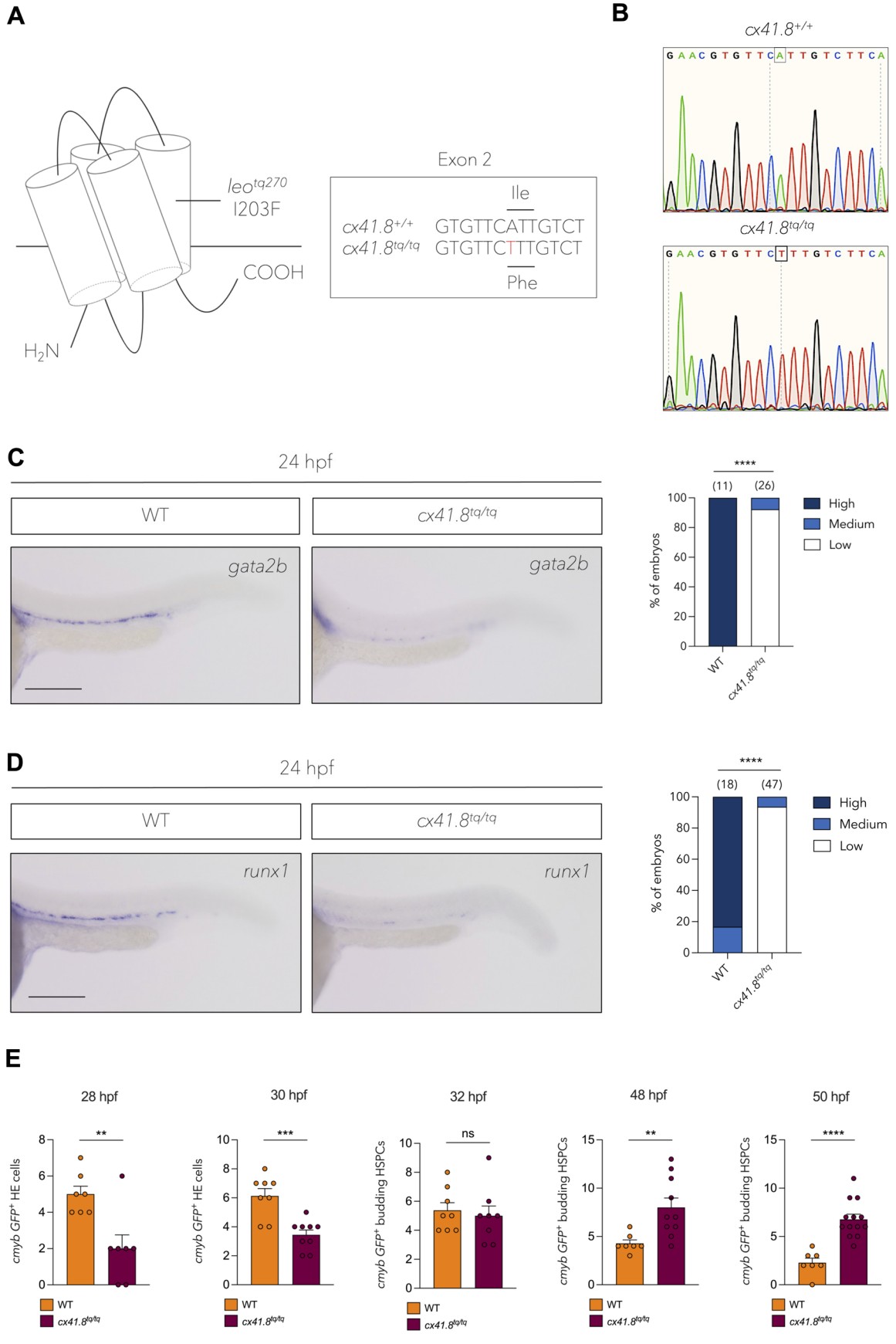

**Fig. 1.** See next page for legend.

**Fig. 1. The I203F mutation in Cx41.8 results in a defect in haemogenic endothelium induction and HSPC specification.** (A) The *leo^tq270^* (*cx41.8^tq/tq^*) mutant possesses an I203F change in the fourth transmembrane domain. (B) Sanger sequencing shows an A-to-T base change in the *cx41.8^tq/tq^* mutant. (C) *In situ* hybridisation and quantification of *gata2b* in *cx41.8^tq/tq^* mutants and controls at 24 hpf. (D) *In situ* hybridisation and quantification of *runx1* in *cx41.8^tq/tq^* mutants and controls at 24 hpf. (E) Quantification of *cmyb:GFP^+^* haemogenic endothelial (HE) cells and HSPCs in *cx41.8^tq/tq^* and control embryos between 28 and 50 hpf. Statistical significance was calculated using either a Chi-squared test (C and D) or an unpaired *t*-test (E). *P<0.05, **P<0.01, ***P<0.001, ****P<0.0001. Scale bars: 200 μm (C and D). Created in BioRender by Petzold, T., 2025. https://BioRender.com/97nw7ib. This figure was sublicensed under CC-BY 4.0 terms.

Interestingly, however, there was no difference in the number of *cmyb^+^* HSPCs budding from the dorsal aorta between *cx41.8^tq/tq^* mutants and controls at 32 hpf (Fig. 1E), whilst at 48 and 50 hpf, an increase in the number of *cmyb:GFP^+^* budding HSPCs was present in *cx41.8^tq/tq^* embryos, compared to wild-type controls (Fig. 1E), pointing towards an HSPC specification delay in these animals. Of note, we also found a reduction in the number of *cmyb:GFP^+^* multiciliated cells along the yolk tube extension of *cx41.8^tq/tq^;cmyb:GFP* embryos relative to controls between 28 and 50 hpf (Fig. S3A,B), although this particular finding was not investigated further.

Since HSPCs are specified from the dorsal aorta in vertebrates (Clements and Traver, 2013), we asked whether arterial EC specification was impaired in *cx41.8^tq/tq^* embryos by analysing *dll4* expression. However, *dll4* expression was normal in *cx41.8^tq/tq^* embryos, as determined by WISH at 24 hpf (Fig. S1C) and 28 hpf (Fig. S1D). Altogether, this data indicates that *cx41.8* plays a role in the induction of the haemogenic endothelium and subsequent specification of HSPCs from the dorsal aorta.

### The HSPC specification defect in *cx41.8^tq/tq^* mutants is due to a delay in *gata2b* expression

To further characterise the HSPC specification defect present in *cx41.8^tq/tq^* mutants, we carried out WISH for the HSPC marker genes *runx1* and *cmyb* at later stages in development. WISH staining marking HSPCs was found to be normal in *cx41.8^tq/tq^* mutant embryos at 48 hpf (Fig. S2B), 72 hpf (Fig. S2C) and 4.5 dpf (Fig. S2D) as determined by *cmyb* WISH.

Hence, since HSPC specification is initially reduced, but then recovers in *cx41.8^tq/tq^* embryos, we suspected a delay in the formation of the haemogenic endothelium in these mutants. To test this hypothesis, we determined the expression of *gata2b* at multiple stages in *cx41.8^tq/tq^* embryos, since its expression precedes the expression of *runx1*, and marks the development of the haemogenic endothelium (Butko et al., 2015). *gata2b* was found to be significantly reduced at 23 hpf (Fig. S4A), 24 hpf (Fig. S4B) and 26 hpf (Fig. S4C) in *cx41.8^tq/tq^* mutants compared to control embryos. However, *cx41.8^tq/tq^* embryos displayed significantly more *gata2b* expression at 30 hpf, 32 hpf and 36 hpf (Fig. S5A-C). To validate our *in situ* hybridisation data findings, we also performed gene expression analysis of *gata2b* (Fig. S6A), *runx1* (Fig. S6B) and *cmyb* (Fig. S6C) by qPCR on dissected trunks and tails of *cx41.8^tq/tq^* embryos and controls at different developmental stages. Broadly, our qPCR data corroborates our WISH findings. Notably, however, the gene expression differences observed by qPCR were not found to be statistically significant, which may be explained by the relatively high variation in expression present between replicates.

Altogether, however, these data demonstrate delayed *gata2b* expression in the dorsal aorta of *cx41.8^tq/tq^* embryos, which we speculate also results in the delay in the expression induction of the

downstream genes, *runx1* and *cmyb*. As such, *cx41.8^tq/tq^* embryos possess a developmental delay in the formation of the haemogenic endothelium, which subsequently results in the disrupted temporal control of HSPC specification.

### *cx41.8* is expressed in ECs in the dorsal aorta floor

We previously showed that *cx41.8* is expressed in vascular ECs in zebrafish (Denis et al., 2019), and we also reported a critical role for Cx41.8 in bridging HSPCs with their vascular niche in the caudal haematopoietic tissue (Cacialli et al., 2021). Additionally, transcriptomics data recently revealed a high expression of *cx41.8* in arterial ECs at 24 hpf (Gurung et al., 2022). In order to investigate the spatiotemporal expression pattern of *cx41.8* more precisely, we generated a *cx41.8:EGFP* zebrafish reporter line using the previously described *cx41.8* promoter (Watanabe and Kondo, 2012) (Fig. 2A).

During HSPC specification (24-48 hpf), *EGFP* was expressed in structures resembling vasculature. In particular, strong expression was detected as a thin line in the region of the axial vasculature (likely the aortic floor, see 24 and 28 hpf in Fig. 2B). Furthermore, at 48 hpf, *cx41.8:EGFP* expression was detected in rounded cells at the floor of the aorta, which are presumptive budding HSPCs (Fig. 2B). To confirm these findings, we established *cx41.8:EGFP; kdrl:mCherry* double transgenic embryos. The trunks and tails were dissected from these embryos at 48 hpf and subjected to flow cytometry analyses. A population of double positive *cx41.8:EGFP; kdrl:mCherry* cells was detected, which was absent in *kdrl:mCherry* embryos (Fig. 2C, for gating strategy see Fig. S7). This confirms that *cx41.8* is expressed in ECs of the zebrafish trunk and tail.

Finally, by observing *cx41.8:EGFP;kdrl:mCherry* embryos by fluorescence microscopy, we confirmed that double positive *cx41.8:EGFP;kdrl:mCherry* ECs are indeed present in the floor of the aorta between 24 and 48 hpf (Fig. 2D). Together, this suggests that *cx41.8* is expressed in presumptive haemogenic ECs and budding HSPCs in the aortic floor during the time of EHT and HSPC specification.

### Temporal mitochondrial ROS induction of HSPCs requires *cx41.8*

Previous work has elucidated a key role of mitochondrial-derived ROS in HSPC specification (Harris et al., 2013). Furthermore, the mammalian Cx41.8 orthologue, CX40, has been found to be localised to the mitochondria in mouse and human ECs, and was shown to be necessary for mitochondrial ROS production (Guo et al., 2017). Hence, we wanted to test if the production of mitochondrial ROS was impaired in ECs of the vascular cord, which begins to luminise around 18 hpf (Jin et al., 2005), in *cx41.8^tq/tq^* mutants. To determine whether this was the case, we first probed for the presence of total cellular ROS and mitochondrial-derived ROS in the vascular cord at 16 hpf (Fig. 3A). Total cellular ROS were detected in the ventral side of the vascular cord in *kdrl:GFP* embryos, but were absent in *cx41.8^tq/tq^;kdrl:GFP* embryos, as determined by using a cellROX probe (Fig. 3B). Similarly, mitochondrial ROS were also detected on the ventral side of the vascular cord in *kdrl:GFP* embryos at 16 hpf, but were absent in *cx41.8^tq/tq^;kdrl:GFP* embryos, as determined using a mitoSOX probe (Fig. 3C). This indicates that there is indeed a defect in mitochondrial ROS production in vascular cord ECs in *cx41.8^tq/tq^* mutant embryos, prior to the induction of *gata2b* expression.

Next, we set out to determine whether modulation of ROS has an effect on haemogenic endothelium induction and HSPC specification. Wild-type (WT) embryos were treated with MitoTEMPO, a specific inhibitor of mitochondrial ROS production (Peterman et al., 2015),

**A**

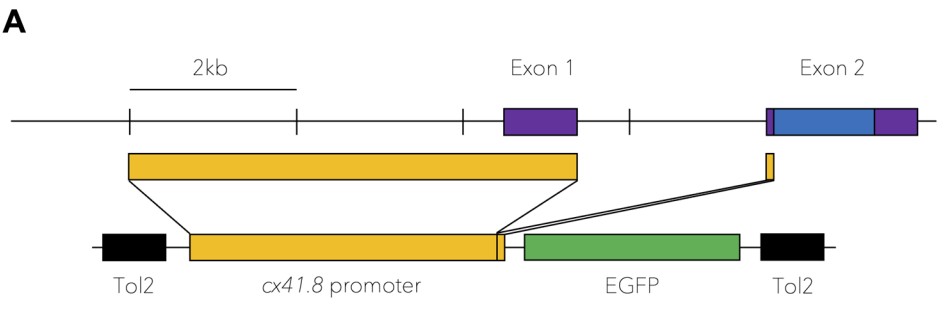

**B**

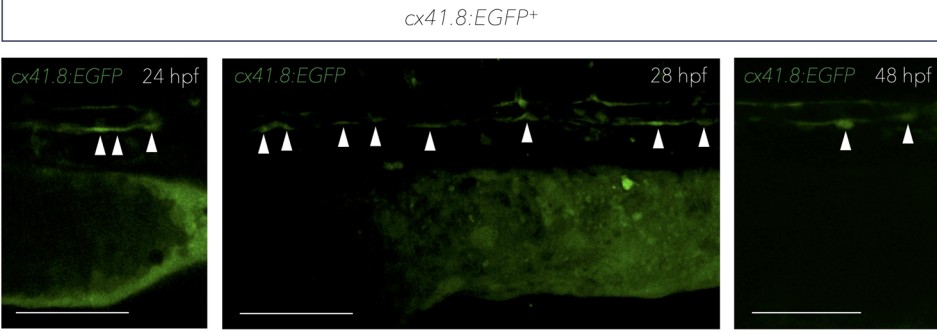

**Fig. 2. *cx41.8* is expressed in presumptive haemogenic endothelial cells of the dorsal aorta**. (A) Design of the *cx41.8:EGFP* plasmid. The upper line indicates the *cx41.8* locus structure. The lower line indicates the construct design. Purple boxes indicate the *cx41.8* exons; blue boxes indicate the open reading frame; yellow boxes indicate the 4.5 kb sequence upstream of the *cx41.8* start codon; black boxes indicate the transposon sequences and the green box indicates the EGFP coding sequence. (B) *cx41.8:EGFP* expression in the presumptive floor of the aorta at 24 and 28 hpf and in presumptive budding HSPCs (48 hpf). White arrowheads denote presumptive haemogenic endothelial cells in the floor of the aorta and budding HSPCs (48 hpf). (C) Flow cytometry analysis of double-positive cells in 48 hpf *kdrl:mCherry*⁺ or *cx41.8:EGFP*⁺; *kdrl:mCherry*⁺ embryos. (D) Expression of *cx41.8:EGFP* and *kdrl:mCherry* from 24-48 hpf. White arrowheads denote *cx41.8:EGFP* and *kdrl:mCherry* double-positive endothelial cells in the floor of the dorsal aorta. Scale bars: 100 μm (B and D).

**C**

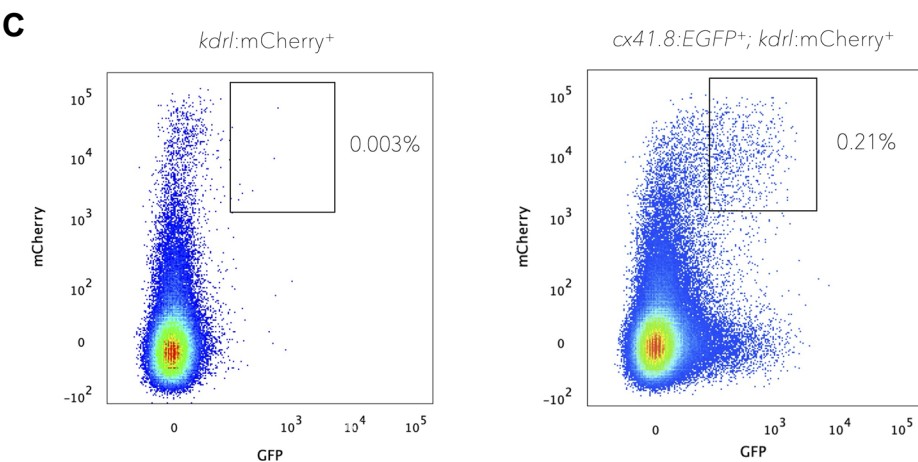

**D**

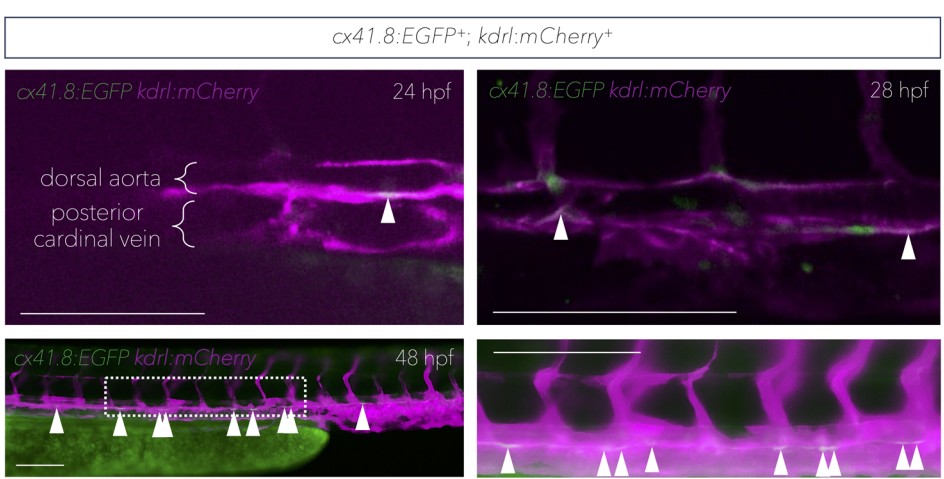

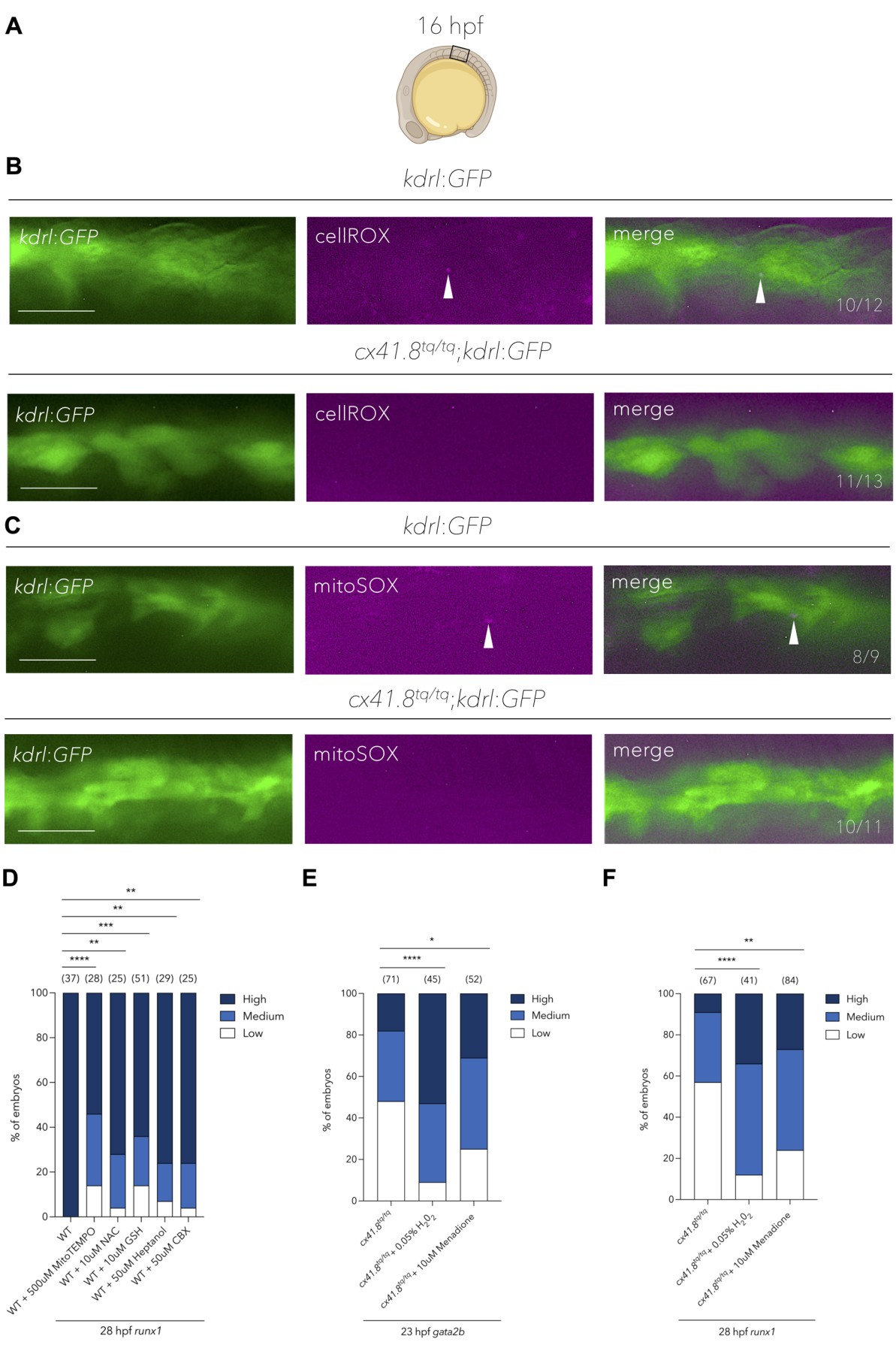

**Fig. 3.** See next page for legend.

**Fig. 3. Mitochondrial-derived ROS production in endothelial cells is required for haemogenic endothelium induction and the specification of HSPCs.** (A) Schematic showing the region of 16 hpf embryos which was analysed by fluorescence microscopy in B and C. (B) Fluorescence microscopy images of total cellular ROS detection in *kdrl*:GFP⁺ or *cx41.8*^(tq/tq);*kdrl*:GFP⁺ embryos. White arrowheads denote the presence of ROS in endothelial cells. Numbers indicate the ratio of embryos with the respective phenotype. (C) Fluorescence microscopy images of mitochondrial-derived ROS detection in *kdrl*:GFP⁺ or *cx41.8*^(tq/tq);*kdrl*:GFP⁺ embryos. White arrowheads denote mitochondrial-derived ROS in endothelial cells. Numbers indicate the ratio of embryos with the respective phenotype. (D) Quantification of aortic *runx1* signal (*in situ* hybridisation) at 28 hpf in WT control embryos and those treated with MitoTEMPO, NAC, GSH, heptanol or CBX. (E) Quantification of aortic *gata2b* signal (*in situ* hybridisation) at 23 hpf in *cx41.8*^(tq/tq) control embryos and *cx41.8*^(tq/tq) embryos treated with $H_2O_2$ or menadione. (F) Quantification of aortic *runx1* signal (*in situ* hybridisation) at 28 hpf in *cx41.8*^(tq/tq) control embryos and *cx41.8*^(tq/tq) embryos supplemented with $H_2O_2$ and menadione. Statistical significance was calculated using a Chi-squared test. *$P<0.05$, **$P<0.01$, ***$P<0.001$, ****$P<0.0001$. Scale bars: 25 µm (B and C). Created in BioRender by Petzold, T., 2025. https://BioRender.com/n88jg67. This figure was sublicensed under CC-BY 4.0 terms.

and subsequently fixed for WISH. Treatment with MitoTEMPO from 14 hpf resulted in impaired HSPC specification as determined by a significant reduction of *runx1* WISH signal at 28 hpf (Fig. 3D). Treatment of WT embryos with the anti-oxidants N-Acetyl-Cysteine (NAC) or reduced L-glutathione (GSH) from 14 hpf, also impaired HSPC specification as determined by *runx1* WISH at 28 hpf (Fig. 3D), as was previously shown (Harris et al., 2013). Moreover, treatment of WT embryos with the connexin blockers heptanol (Muto and Kawakami, 2011) and carbenoxolone (CBX) (Casano et al., 2016) from 14 hpf also resulted in a decrease in HSPC specification (Fig. 3D).

Following this, we treated *cx41.8*^(tq/tq) mutant embryos with the ROS enhancers, $H_2O_2$ (Zorov et al., 2014) and menadione (Criddle et al., 2006) from 14 hpf, which resulted in an increased expression of both *gata2b* at 23 hpf (Fig. 3E) and *runx1* at 28 hpf (Fig. 3F), as determined by WISH. $H_2O_2$ was able to rescue *gata2b* expression in a dose-dependent manner (Fig. S8). Together, this data suggests that Cx41.8 plays a key role for the correct temporal generation of mitochondrial ROS and the subsequent induction of the haemogenic program in the dorsal aorta.

### Induction of the Hif1/2α-mediated haematopoietic program in response to mitochondrial ROS is dependent on *cx41.8*

Recent research has demonstrated that hypoxia and mitochondrial ROS are required for the stabilisation of the transcription factors Hif1/2α at the protein level (Harris et al., 2013; Gerri et al., 2018). Mechanistically, ROS stabilise Hif1/2α by inhibiting prolyl hydroxylases which target Hif1/2α for ubiquitination by the von Hippel Lindau (VHL) protein, resulting in their subsequent degradation (Pan et al., 2007; Chowdhury et al., 2016). Hif1/2α have been shown to act upstream of Notch1a/b signalling, which in turn induces *gata2b* expression and the formation of the haemogenic endothelium (Gerri et al., 2018). Therefore, we hypothesised that the lack of mitochondrial ROS production at 16 hpf in *cx41.8*^(tq/tq) embryos resulted in the degradation of Hif1/2α and the subsequent lack of *gata2b* transcriptional activation.

To test whether *cx41.8* is involved in this ROS-Hif1/2α-Notch1a/b-*gata2b* pathway, *cx41.8*^(tq/tq) embryos were treated with either cobalt chloride (CoCl₂), a hypoxia mimetic that interferes with the interaction between Hif1/2α and VHL (Yuan et al., 2003), or the prolyl hydroxylase inhibitor dimethyloxallyl glycine (DMOG) (Takeda et al., 2009). Treatment with CoCl₂ or DMOG from 14 hpf

resulted in a rescue of both *gata2b* (Fig. 4A) and *runx1* (Fig. 4B) expression in *cx41.8*^(tq/tq) embryos at 23 and 28 hpf, respectively.

We then used a previously described *vhl* morpholino (MO) (Klems et al., 2020; Wild et al., 2017; Santhakumar et al., 2012) to prevent *vhl* function. The *vhl*-MO resulted in the induction of a cryptic splice site in exon 1 of the *vhl* transcript (Fig. S9A) and the subsequent loss of 18 amino acids from the VHL beta domain (Fig. S9B), required for the interaction between VHL and Hif1/2α (Haase, 2009). MO-mediated knockdown of *vhl* in WT embryos resulted in an increase in *gata2b* expression at 23 hpf (Fig. 4C) whilst there was a non-significant increase in *runx1* expression at 28 hpf (Fig. 4D), since expression of this marker is already high in the majority of control MO-injected WT embryos. MO-mediated knockdown of *vhl* in *cx41.8*^(tq/tq) embryos resulted in a rescue of *gata2b* expression at 23 hpf (Fig. 4C) and also rescued *runx1* expression at 28 hpf (Fig. 4D), showing that the delay in *gata2b* expression was likely due to the degradation of Hif1/2α.

Finally, in order to solidify our model, we performed qPCR on dissected trunks and tails of *cx41.8*^(tq/tq) embryos and controls to investigate the expression of the *hif2a* and *notch1* paralogues, *hif2aa* and *notch1b*. While we expected the expression of *hif2aa* to be similar in *cx41.8*^(tq/tq) embryos and controls, we predicted a delay in the expression of *notch1b* in *cx41.8*^(tq/tq) mutants. Indeed, we found that while *hif2aa* expression levels remained relatively equal between *cx41.8*^(tq/tq) embryos and controls between 24 and 36 hpf (Fig. S10A), *notch1b* expression was reduced in *cx41.8*^(tq/tq) embryos at 24 and 36 hpf (Fig. S10B), which was no longer the case at 48 hpf (Fig. S10B). In summary, this data provides evidence for the involvement of *cx41.8* in a ROS-Hif1/2α-Notch1a/b-*gata2b* pathway that governs timely induction of the haemogenic endothelium and HSPC specification.

### DISCUSSION

Here, we have demonstrated that there is a delay in the development of the haemogenic endothelium and the subsequent specification of HSPCs in the absence of a fully functional Cx41.8. Mechanistically, Cx41.8 seems to play a key role in the production of mitochondrial ROS. This may be the result of reduced calcium uptake into the mitochondria in *cx41.8*^(tq/tq) mutants, as has been reported in *CX40*-deficient ECs (Guo et al., 2017). Furthermore, whilst an increase in mitochondrial metabolism was previously found to drive mitochondrial ROS generation in response to glucose, enhancing HSPC induction (Harris et al., 2013), calcium entry into the mitochondria has also been demonstrated to induce mitochondrial metabolism (Rossi et al., 2019; Jouaville et al., 1999). Hence, we speculate that mitochondrial calcium influx, occurring in a Cx41.8 dependent manner, drives mitochondrial metabolism, resulting in an increase in mitochondrial ROS production. Ultimately, this elevation in ROS then induces the downstream haematopoietic program.

The partial functionality of the Cx41.8 channel in *cx41.8*^(tq/tq) mutants (Watanabe et al., 2006) may explain why the HSPC program is delayed but induced, as mitochondrial ROS generation may eventually reach the threshold required to sufficiently stabilise the Hif1/2α proteins for downstream transcriptional activation of *gata2b*. This could also result from functional redundancy between Cx41.8 and other connexins such as Cx43 or Cx45.6 in the mitochondria, since they are also expressed in zebrafish arterial ECs at 24 hpf (Gurung et al., 2022) and *cx43* knockdown has previously been shown to result in an HSPC specification defect in zebrafish (Jiang et al., 2010). This potential functional redundancy may also provide an explanation as to why HSPCs are specified normally, without any delay, in *cx41.8*^(t1/t1) embryos (Cacialli et al., 2021). In

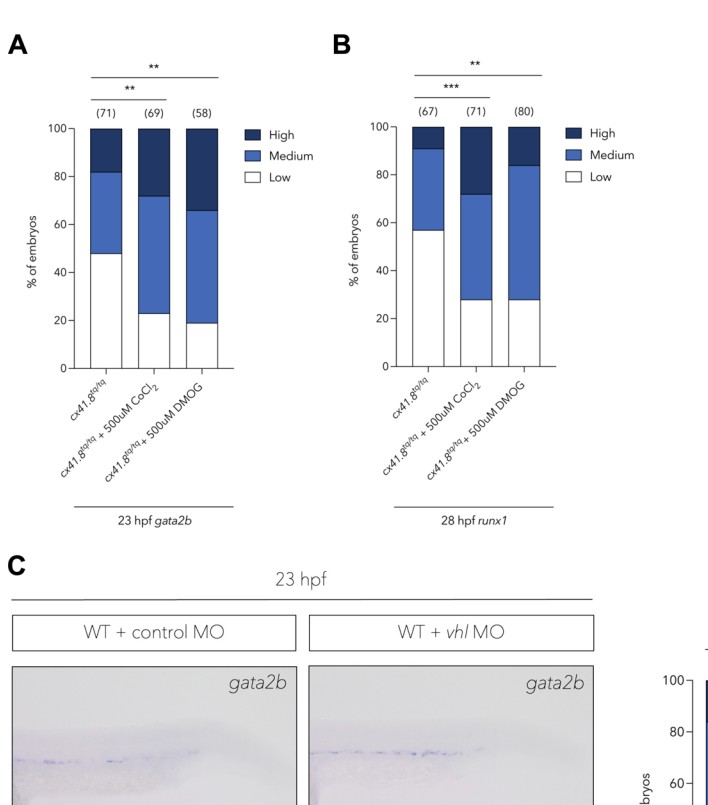

**A** 23 hpf *gata2b*

**B** 28 hpf *runx1*

**Fig. 4. Stabilisation of Hif1/2α rescues haemogenic endothelium induction and HSPC specification in *cx41.8<sup>tq/tq</sup>* mutants.** (A) Quantification of aortic *gata2b* signal (*in situ* hybridisation) at 23 hpf in *cx41.8<sup>tq/tq</sup>* control embryos and *cx41.8<sup>tq/tq</sup>* embryos supplemented with CoCl₂ or DMOG. (B) Quantification of aortic *runx1* signal (*in situ* hybridisation) at 28 hpf in *cx41.8<sup>tq/tq</sup>* control embryos and *cx41.8<sup>tq/tq</sup>* embryos treated with CoCl₂ or DMOG. (C) *In situ* hybridisation and quantification of *gata2b* at 23 hpf in WT control embryos and *cx41.8<sup>tq/tq</sup>* embryos injected with either control- or *vhl*-MO. (D) *In situ* hybridisation and quantification of *runx1* at 28 hpf in control-MO or *vhl*-MO injected wild-type or *cx41.8<sup>tq/tq</sup>* mutant embryos. Statistical significance was calculated using a Chi-squared test. \*$P<0.05$, \*\*$P<0.01$, \*\*\*$P<0.001$, \*\*\*\*$P<0.0001$. Scale bars: C, 100 µm, D, 200 µm.

**C** 23 hpf

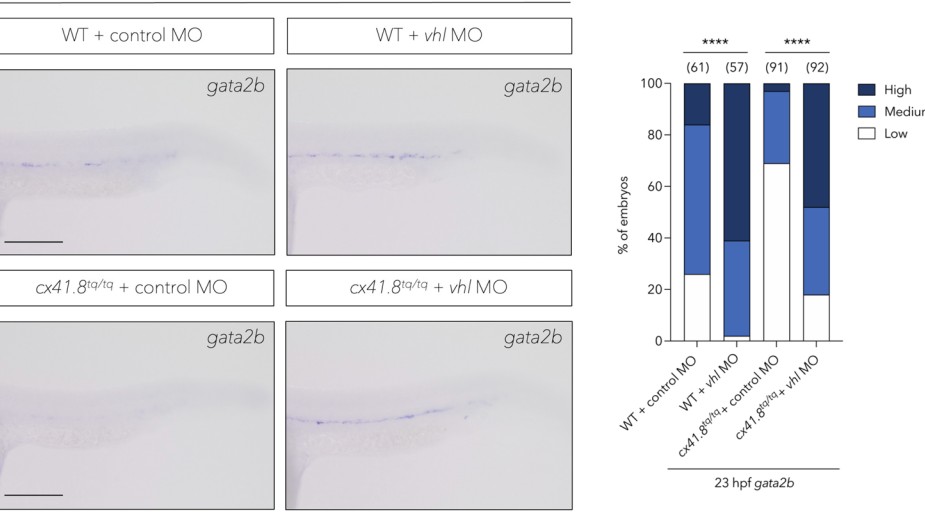

23 hpf *gata2b*

**D** 28 hpf

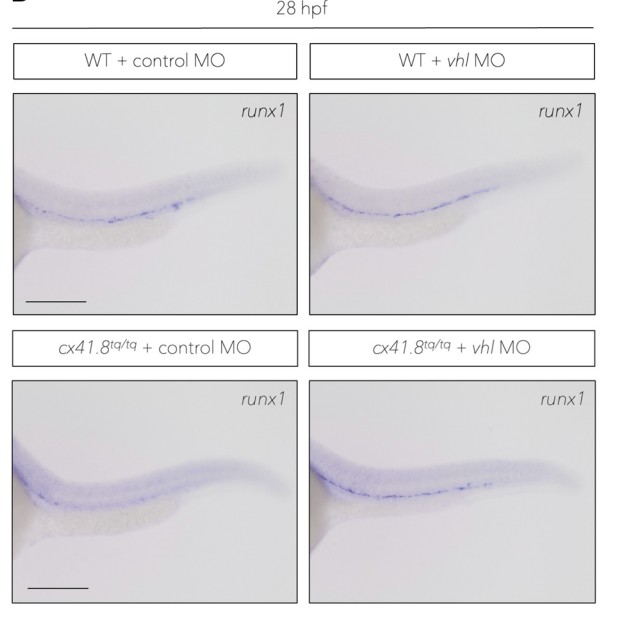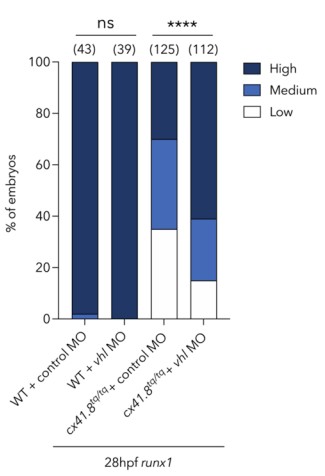

28hpf *runx1*

these null mutants, *cx41.8* expression is completely absent but may be functionally compensated by other connexins, whereas in *cx41.8^{tq/tq}* mutants, although *cx41.8* is expressed, its channel function is reduced (Watanabe et al., 2006, 2016). Moreover, as Cx41.8 may form heterotypic channels with Cx43 and/or Cx45.6 (and potentially also with others), the function of these chimeric channels would also be altered.

GATA2 has been shown to positively autoregulate its own expression in mice (Katsumura et al., 2016; Nozawa et al., 2009), and Gata2b may also act in this way in zebrafish (Dobrzycki et al., 2020). Therefore, one can speculate that once *gata2b* expression has been induced by the Cx41.8-mitoROS-Hif1/2α-Notch1a/b-*gata2b* pathway, it may also further activate its own expression, increasing robustness of the haematopoietic transcriptional program. In any case, whether *gata2a*, a paralogue of *gata2b*, which was previously shown to contribute to definitive haematopoiesis in zebrafish (Gioacchino et al., 2021; Bresciani et al., 2021), also plays a role in the molecular pathway that we have uncovered, remains to be explored in future work.

In summary, we suggest that mitochondrial channels formed by Cx41.8 and perhaps also others, are important for ROS production in the mitochondria of vascular cord ECs, as early as 16 hpf. These mitochondrial-derived ROS stabilise the transcription factors Hif1/2α, which subsequently translocate into the nucleus leading to the expression induction of *gata2b* in a Notch1a/b-dependent manner (Fig. 5), as shown previously (Gerri et al., 2018). This signalling cascade ends with the specification of the haemogenic endothelium, leading to the formation of HSPCs. *cx41.8^{tq/tq}* mutants display a delay in the transcriptional program regulating the formation of the haemogenic endothelium and HSPCs. Hence, *cx41.8* facilitates

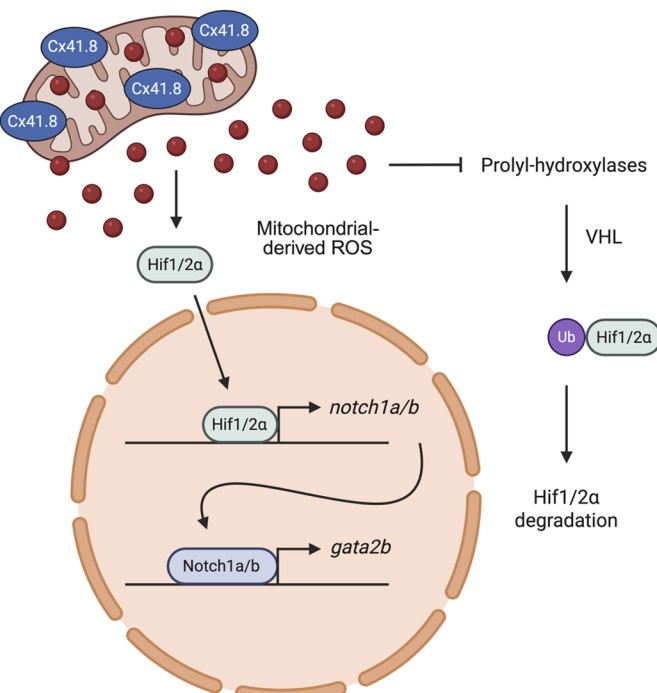

**Haemogenic endothelial cell**

**Fig. 5. Proposed model of the role of Cx41.8 in HSPC specification.** Cx41.8 localises to the mitochondria in haemogenic endothelial cells, allowing mitochondrial ROS production which stabilises Hif1/2α, which in turn induces *gata2b* expression via Notch1a/b signalling. Created in BioRender by Petzold, T., 2025. https://BioRender.com/konhezh. This figure was sublicensed under CC-BY 4.0 terms.

the correct temporal induction of HSPCs. We speculate that this is likely to also be the case in mammals, since *CX40* (the mammalian orthologue of *cx41.8*) is highly expressed in the haemogenic endothelium of mouse (Fadlullah et al., 2022) and humans (Calvanese et al., 2022), and as CX40 localises to the mitochondria in ECs of both these species (Guo et al., 2017).

Further research will be required to determine the exact mechanism(s) by which mitochondrial ROS are produced specifically in this EC subpopulation. This may involve sterile inflammation, which is particularly important for the induction of EHT and the budding of nascent HSPCs from the aortic floor (Espin-Palazon et al., 2014; Li et al., 2014; Collins et al., 2021). Our data contribute to a better understanding of the factors required for the initiation of the haemogenic program, which may have significant implications for enhancement of current regenerative medicine protocols, to produce haematopoietic progenitor cells *in vitro*.

## MATERIALS AND METHODS

### Ethical statement
Zebrafish were raised in accordance with FELASA and Swiss guidelines (Alestrom et al., 2020). No authorisation was required since experiments were carried out on embryos up to 5 days post fertilization. All efforts were made to comply to the 3R guidelines.

### Zebrafish husbandry
AB* zebrafish, as well as transgenic zebrafish lines were kept in a 14/10 h light/dark cycle at 28.5°C. Embryos were obtained as described previously (Westerfield, 2000). Embryos were staged by hours post fertilization (hpf) as described previously (Kimmel et al., 1995). In this study the mutant zebrafish line *leo^{270/270}* (Watanabe et al., 2006) (referred to as *cx41.8^{tq/tq}*) was utilised and genotyped by PCR (using primers listed in Table S1), followed by Sanger sequencing (Fig. 1B). The following transgenic lines were used in this study: *Tg(kdrl:GFP)^{s843}* (Jin et al., 2005), *Tg(kdrl:Has.HRASmCherry)^{s896}* (Chi et al., 2008) (referred to as *Tg(kdrl:mCherry))*, *Tg(cmyb:GFP)^{zf169}* (North et al., 2007) and *Tg(cx41.8:EGFP)* (Watanabe and Kondo, 2012). Zebrafish embryos were treated with 0.003% 1-phenyl-2-thiourea (PTU, Sigma P7629) starting at 24 hpf to prevent pigmentation.

### Generation of transgenic animals
For *Tg(cx41.8:EGFP)* zebrafish generation, 50 pg of the Tol2 *cx41.8:EGFP* plasmid, described previously (Watanabe and Kondo, 2012), was co-injected with 50 pg of *tol2 transposase* mRNA into AB* zebrafish embryos. Injected F0s were mated with AB* zebrafish, and the resulting F1 offspring were screened by fluorescence microscopy to assess germline integration of the Tol2 construct. F2 zebrafish adults were subsequently mated and their offspring utilised in experiments.

### WISH
WISH was performed on 4% paraformaldehyde-fixed embryos as described previously (Thisse and Thisse, 2008). Digoxigenin-labelled *dll4*, *gata1*, *pu.1*, *gata2b*, *runx1* and *cmyb in situ* probes were used and their generation has been described previously (Cacialli et al., 2021; Mahony et al., 2016).

### Chemical treatments
All compounds used in this study were purchased from Sigma-Aldrich. Zebrafish embryos were exposed to compounds in 0.003% 1-phenyl-2-thiourea (PTU, Sigma, P7629) E3 (fish) water in multi-well plates from 14 hpf to either 23 or 28 hpf. Following exposure, embryos were fixed in 4% paraformaldehyde. All chemical treatment experiments are a combination of at least two independent experiments with independent clutches.

### Flow cytometry
Dissected embryos were incubated with a liberase-blendzyme 3 (Roche) solution for 90 min at 33°C, then dissociated and resuspended in 0.9× PBS-1% fetal calf serum, as described previously (Cacialli et al., 2021). We distinguished and excluded dead cells by staining them with SYTOX Red

(Life Technologies). Cell suspensions were passed through a 40 mm filter prior to flow cytometry. Data were acquired on a LSR2Fortessa (BD Biosciences, software diva8.0.2) and analysed with FlowJo (v10).

### Total cellular and mitochondrial reactive oxygen species detection

Whole-mount staining with the cellROX (Invitrogen) or MitoSOX (Life Technologies) probes was performed on living zebrafish embryos at 16 hpf, following the methods described previously (Mugoni et al., 2014). Embryos were exposed to either a 5 µM cellROX or a 5 µM MitoSOX solution for 45 min and incubated at 28.5°C. Subsequently, imaging was carried out by fluorescence microscopy.

### Microscopy

WISH images were taken on an Olympus MVX10 microscope in 100% glycerol. Confocal imaging in Fig. 2B and D was performed using an upright 3i spinning-disc confocal microscope and using a Zeiss Plan-Apochromat 20× or 40× water-dipping objective. All other fluorescent images were taken with an IX83 microscope [Olympus; Figs 2D (48 hpf), 3B,C and Fig. S3A]. All images were taken using the CellSens Dimension software (Olympus).

### MO injections

The standard control (CCTCTTACCTCAGTTACAATTTATA) and *vhl* (GCATAATTTCACGAACCCACAAAAG) MO oligonucleotides were purchased from GeneTools (Philomath, OR, USA). MO efficiency was tested by PCR (Fig. S9C) from total RNA extracted from 15 embryos per sample at 24 hpf (using primers listed in Table S3). The *vhl*-MO-induced loss of 54 bp from exon 1 of the *vhl* transcript upon injection of 6 ng of MO was confirmed by Sanger sequencing (Fig. S9D). In all subsequent MO experiments, 6 ng of *vhl* or standard control MO was injected per embryo. All morpholino experiments are a combination of three experiments with independent clutches.

### Quantitative real-time PCR and analyses

Total RNA was extracted using RNeasy minikit (Qiagen) and reverse transcribed into cDNA using a Superscript III kit (Invitrogen). Quantitative real-time PCR (qPCR) was performed using a KAPA SYBR FAST Universal qPCR Kit (KAPA BIOSYSTEMS) and run on a CFX connect real-time system (Bio-Rad). All qPCR primers used for gene expression are listed in Table S2. Trunks and tails of embryos were dissected from embryos for all qPCR experiments and ∼30 embryos were used for each condition. All qPCR experiments were performed using technical triplicates. All expression values were normalised to the expression of *ef1-alpha*. Experiments were each repeated two or three times and fold-change averages from each experiment were combined.

### Image processing and analyses

All images were processed using Fiji ImageJ (NIH) (Schindelin et al. 2012). For quantification of *cmyb* $GFP^+$ haemogenic endothelium, HSPCs and multi-ciliated cells, cells in the trunk region spanning the length of the yolk tube extension were included in the analysis in each case. WISH phenotypic variation was analysed qualitatively and depicted graphically as the percentage of total embryos scored exhibiting high, medium or low gene expression in the region of interest, as done previously (Lefkopoulos et al. 2020). In detail, qualitative scoring (number of embryos with altered signal per total number of embryos making up the sample for each case) of WISH staining was conducted manually. The individual performing the categorization evaluated the staining intensity and, due to the versatility of staining from experiment to experiment, used the staining exhibited by the majority of the control embryos of each individual experiment as a reference point. The individual categorised as 'low expression' the embryos depicting a staining lower than the staining of the majority of control embryos, as 'medium expression' the embryos depicting a staining approximately equal to the staining of the majority of control embryos and as 'high expression' the embryos showing a higher intensity compared to the staining of the majority of control embryos. The *in situ* hybridization scoring presented in the manuscript represents the work of one individual, to avoid combining

systematic errors. The individual categorizing the phenotypes into low, medium, high turned the numbers of embryos into % percentages (number of embryos depicting the particular staining intensity/total number of embryos examined in the experimental sample×100%), before statistical analyses was carried out. Details of the statistical test(s) used to determine significance are described in the section Data analyses.

### Data analyses

At least three independent experiments were carried out in all cases, unless stated otherwise. In all experiments, normality was assumed, and variance was comparable between groups. Sample size was selected empirically according to previous experience in the assessment of experimental variability and sample sizes are indicated in each figure, when appropriate. Numerical data are the mean±s.e.m., unless stated otherwise. Statistical tests used are stated in each figure legend. Statistical differences are denoted as *$P<0.05$, **$P<0.01$, ***$P<0.001$, ****$P<0.0001$ in all figures. Statistical calculations and the graphs for the numerical data were performed using Prism 10 software (GraphPad Software).

### Acknowledgements
We would like to thank all lab members for their comments and suggestions during this project. We would also like to thank Prof. Brenda Kwak for helpful discussions about connexins. This work would not have been possible without support from the animal, flow-cytometry and imaging facilities at The University of Geneva.

### Competing interests
The authors declare no competing or financial interests.

### Author contributions
Conceptualization: J.Y.B.; Data curation: T.P., S.B.; Formal analysis: T.P., S.B., T.L.; Methodology: T.P., J.Y.B.; Resources: M.W., H.G.; Supervision: J.Y.B.; Writing – original draft: T.P., J.Y.B.; Writing – review & editing: T.P., J.Y.B.

### Funding
T.P. benefitted from a grant from the Gabbiani Fund. J.Y.B. was funded by the Swiss National Fund (grant #310030_184814) and by the Fondation Privée des Hopitaux de Genève. Open Access funding provided by University of Geneva. Deposited in PMC for immediate release.

### Data and resource availability
Raw data is accessible on Yareta (https://yareta.unige.ch/archives/74e35492-529f-4266-a898-4f43dc5e6102) and in the supplementary information.

### Peer review history
The peer review history is available online at https://journals.biologists.com/bio/lookup/doi/10.1242/bio.062118.reviewer-comments.pdf.

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
