## [Peer Review File · Biology Open]

Connexin 41.8 governs timely haematopoietic stem and progenitor cell specification

Tim Petzold, Sarah, Brivio, Tanja, Linnerz, Masakatsu Watanabe, Holger Gerhardt, Julien Y. Bertrand

DOI: 10.1242/bio.062118

Editor: Tristan Rodriguez

Review timeline

Submission to Review Commons:	30 August 2023
Original submission to affiliate journal:	6 November 2023
Editorial decision:	15 November 2024
Revision received:	3 April 2025
Revision decision:	13 June 2025
Transfer to Biology Open	16 June 2025
Accepted:	18 June 2025

Reviewer 1:

Evidence, reproducibility and clarity

Summary

The manuscript by Petzold et al. explores the functions of connexin 41.8 (cx41.8) (mammalian homologue Connexin 40) in hematopoietic stem cell (HSC) formation in the zebrafish dorsal aorta. The authors use a cx41.8 allele that appears to be hypomorphic, as the phenotype is milder than a previous cx41.8 allele that the same group published (Cacialli et al., 2021). cx41.8tq/tq mutants exhibit delayed onset of hemogenic endothelial specification, as marked by *gata2b* at 24 hpf, but HSPC development proceeds normally from 48 hpf onwards. A new reporter line for cx41.8, Tg(cx41.8:GFP), was generated and is expressed in the floor of the dorsal aorta, consistent with the location of hemogenic endothelial cells. Lower ROS production in the whole cell and in the mitochondria was reported in the cx41.8tq/tq mutants, and treatment with ROS enhancers, H₂O₂ and menadione, appeared to rescue the mutant phenotype of reduced HSPCs at 28 hpf. Finally, the authors tested a link between cx41.8 and Hif1 α by pharmaceutically (DMOG/CoCl₂) or genetically (vhl morpholino) inhibiting Hif inhibitors, and observed a rescue of HSPC formation in cx41.8 mutants.

Major comments

- I think it would be important for the authors to address the mechanisms of why cx41.8tq/tq and the other cx41.8^{-/-} (leot1/t1) mutant phenotypes are different, with the latter allele showing more severe phenotypes of increased HSPC apoptosis and reduced HSPCs during later development. The authors speculate the cx41.8tq/tq allele encodes a missense mutation in one of the channel domains, and as such, might be a hypomorph. The authors cited the original paper by Watanabe et al. (2006); however, this paper actually noted that the cx41.8tq/tq allele is likely to be a dominant negative - and as such, should have exhibited a stronger phenotype than the leot1/t1 mutant allele. From the paper: "leotw28 and leotq270 heterozygotes have phenotypes different from that of WT; thus, they represent dominant- negative alleles." Importantly, no data are shown to provide evidence that the allele is a hypomorph - at minimum, qPCR data should be provided to show whether there is NMD of the mRNA in cx41.8tq/tq mutants.

- One major missing component is experimental data that distinguish the gap junction/plasma membrane- related and the mitochondrial membrane-related functions of Cx41.8. This is critical, as the role of Connexins in the mitochondria remains poorly understood (and Connexin 43 is the best understood one). Thus, it is a big claim by the authors that Cx41.8 primarily acts through the mitochondria and not the gap junctions.
- Suggested experiment: The authors should generate a fluorophore-tagged Cx41.8 - under a ubiquitous (ubb or actin) or HSPC-/hemogenic endothelium-specific (gata2b) promoter to monitor the protein localization of Cx41.8. Providing data on whether Cx41.8 protein indeed localizes to the mitochondria is important to support their claim.
- Related to the above point, the authors should test whether the gap junction function of Cx41.8 is intact in the cx41.8tq/tq mutants by assessing calcium waves in the GCamp transgenic line.
- The quantification data in this manuscript are not satisfactory. The authors only provide graphs that show embryos with "low", "medium" and "high" numbers of HSPCs, which is incredibly subjective. Considering that the authors already have the cx41.8tq/tq in the Tg(myb:GFP) background (Figure 1E), they could have quantified the precise numbers of Tg(myb:GFP)-positive cells at different timepoints and with the different pharmaceutical rescue experiments. Ideally, this should be combined with other HSPC markers such as Tg(cd41:GFP) or Tg(runx1:GFP) - although this could be limited by the authors' access to the lines or time it takes to cross the mutants to the transgenes.
- The link between cx41.8 and Hif1 α is tenuous. The authors should perform in situ hybridization for the hif1 genes and their downstream effector notch1 which is known to be important for the HSPC specification (Gerri et al., 2018). The authors might also want to consider performing transcriptomic analysis (bulk RNA sequencing) from purified HSCs in wild types and cx41.8 mutants and assess the downstream pathways affected by the loss of this gene.

Significance

Overall, this study presents another piece of evidence that Connexin 41.8 regulates HSC formation. It provides a potential link between Connexin 41.8, mitochondrial ROS regulation and Hif/hypoxia-sensitive pathways in promoting endothelial-to-hematopoietic transition. The role of the mitochondrial ROS in particular is quite interesting and might provide a new angle into the role of connexins in regulating hemato-vascular development; however, the authors would need to strengthen the link between Cx41.8 and mitochondrial respiration.

It is important to note that the quantitative data in this manuscript need to be strengthened and refined to strengthen the conclusions. The study is not very deeply mechanistic and appears to be more at an observational/correlational level. The manuscript might be of interest for people in the hematopoietic field but does not shed much more insight into the cellular and molecular mechanisms that govern HSC formation, particularly in light of the paper on Cx41.8 role by the same group (Cacialli et al., 2021).

Reviewer 2:

Evidence, reproducibility and clarity

Summary

Petzold et al are here addressing the potential function of the connexin Cx48.1, a protein involved in the structure of gap junctions, in the specification of future hematopoietic stem cells and progenitors (HSPCs). This piece of work is complementing their previous results showing the function of this connexin isoform in HSPC expansion in the transient hematopoietic niche in the caudal tissue of the zebrafish embryo. They explore phenotypes triggered by the expression of a mutant form bearing a single amino-acid substitution in the fourth transmembrane domain of the protein. Using whole mount in situ hybridization (WISH) of the two transcription factors Gata2b and Runx1, a novel transgenic fish line that expresses eGFP under the control of the Cx48.1 promoter region, and a series of drug treatments interfering with, or promoting, the formation of reactive oxygen species (ROS) production and oxidative stress, they propose that Cx48.1 is also involved upstream of HSPC amplification, rather in their specification at the level

of the hemogenic endothelium constituting the ventral floor of the dorsal aorta. Mechanistically, they hypothesize that this function relies on mitochondria-derived ROS that would destabilize the VHL protein involved in mediating the degradation of Hif1/2a transcription factors, thereby stabilizing the Hif1/2a-Notch1a/b signaling axis involved in specification of the hemogenic endothelium.

****Major comments****

My major comments on the work are on the accuracy of the data in regard to the two main experimental approaches used by the authors and their subsequent analysis/quantification.

1. the WISH and quantitative analyses.

Most of the quantitative analyses in the work are based on chromogenic WISH, which is not sufficiently accurate because leading to highly variable results, in addition to its lack of linearity. WISH is also subjected to important variations, particularly for transcription factors that are expressed at low levels such as Runx1, and to some extent Gata2b also. One obvious example in the paper is the inconsistency of signals that are observed Fig1C (Gata2b, left, wt, 24hpf) and FigS3B (Gata2b, left, wt, 24hpf) in which the signal is barely visible and is comparable to the signal for the cx41.8tq/tq mutant Fig1C, right.

In addition, in the timings that are analyzed in FigS3 (Gata2b, 26 and 28hpf) to argue on temporal delay of expression in the cx41.8tq/tq mutant, the Gata2b signal is masked by the strong increase in tissues other than the hemogenic endothelium in the dorsal aorta (including signal in the somites as well as, possibly, increase in background). In this very example, it is legitimate to question the accuracy of the quantification methodology when the signal in the tissue of interest is drowned in the overall signal from surrounding tissues; how can the authors explain the 100% of embryos that have a 'Low' signal in the region of interest (FigS3C, cx41.8tq/tq mutant in comparison to WT)? This point is also valid for the data quantified FigS4 in which the fitting between WISH data and the quantifications appears to be questionable (for all timing points: 30, 32, 36, 48hpf and comparing mutant with WT). Importantly, it appears also that all over the WISH quantifications, the reader cannot appreciate the accuracy of the categories High/Medium/Low, which is not at all developed in the Methods section (paragraph Image processing and WISH phenotypic analyses); hence, it is not possible to evaluate the accuracy/validity of statistics in particular in the experiments in which the quantification into these categories is used for CoCl2 and morpholino analyses to address the contribution of the Hif1/2a-Notch1a/b pathway Fig4 (these experiments generating results that are not as 'black and white' than the other ones in the paper, hence requiring more accuracy; for example, are the differences in the quantification (% of embryos) significant between the WT+vhl MO and Cx41.8tq/tq mutant + vhl MO? Comparing the 2 WISH results for those conditions does not appear to be very convincing).

My suggestion would be to complement the WISH data and improve the quantitative analyses using another, more accurate approach such as qRT-PCR for example (on dissected trunk regions and, if necessary because of expression in other surrounding tissues (in the case of Gata2b at later time points), after FACS-sorting using a fish line expressing a fluorescent reporter driven by a vascular promoter, ex: the kdrl:mCherry line used in the work). This is particularly important for the expression of the two transcription factors Runx1 and the more upstream Gata2b, the latter being involved in HSPC specification which is taken as a reference. qRT-PCR experiments should be feasible relatively easily and in a reasonable time frame as the technique is not very time consuming and easily accessible.

Finally, there is a confusion in the quantification regarding the number of HSPCs (see the beginning of the second paragraph of Results 'The HSPC specification defect in cx41.8tq/tq mutants is due to a delay in Gata2b expression') and the % of embryos falling into the 3 categories High/Medium/Low FigS2, cmyb 48hpf. The authors use this argument (based on the WISH cmyb signals) to infer that the deficit in the cx41.8tq/tq mutant is not due to controlling HSPC number (no difference in cmyb between WT and mutant) but rather upstream, at the level of the hemogenic endothelium, which is not a thorough argument at that point.

2. Fluorescence imaging and associated interpretation/conclusions.

The fluorescence images (Fig1E; Fig2B,D; Fig3A) are very difficult to analyze; they lack resolution because they appear to be epifluorescence images and not confocal images. When the signal is

low, which is in particular the case for the novel Cx41.8:EGFP fish line, Fig2B (which is confirmed with the FACS GFP signal in comparison to the mCherry of the kdrl:mCherry fish line), it is not possible to provide convincing images on the vascular/aortic expression because of the high background of diffusion (the authors state 'likely to be the aortic floor', indeed it is not possible to validate the fact that the expression is truly in potential hemogenic cells). The double positive population in the FACS (Fig2C, right) does not resolve the issue because if indeed cx41.8 is expressed in endothelial cells (as expected from previous studies), the double positive population could equally be endothelial cells from inter-somitic vessels, for example (not to mention the underlying vein which is very close to the aorta in the trunk)). Fig2D, images are too small and, again, the resolution is not good enough to say that double positive cells are on the aortic floor. It is recommended to convince the reader that the authors try to confirm their statements by using confocal microscopy and increase the magnification of the relevant regions of interest.

There is an inconsistency in the data between Fig1E (40hpf, in vivo imaging using the cmyb:GFP fish line) and FigS2 (48hpf, WISH cmyb); how can we observe 'HSPCs budding from the dorsal aorta' (see legend Fig1, arrowheads) which seems very much decreased in the imaging experiment for the cx41.8tq/tq mutant in comparison to WT, and have no effect on the cmyb signals FigS2B? What are the GFP+ cells that are aligned along the elongated yolk Fig1E and that appeared to be decreased in number in the mutant?

Are the authors sure of their statement on budding HSPCs when the GFP signal pointed by arrows could in majority be hemogenic cells? (which would be in favor of their hypothesis on Cx41.8 being involved rather in hemogenic endothelium/HSPC specification).

****Other Major Comments:****

- It would be important to investigate/show, at least with qualitative WISH experiments all along the time-window of HSPC specification as stated by the authors (26-54hpf, see main text third paragraph of Results), that Cx41.8 is detected in arterial endothelial cells (and perhaps enriched in the hemogenic endothelium?), in complement to the work they are referring to on transcriptomic data at 24hpf (Ref18 Gurung et al Sci Rep 2022). Ideally, these WISH data should be resolute enough to provide clear localization in aortic cells versus cells in the aortic floor to bring significant added value to the work that lacks spatial resolution (ex: fluorescent WISH using confocal microscopy, allowing to superpose signal with cell types (either by double fluorescent WISH (vascular marker + Cx41.8) or superposing fluorescence signals with transmitted light)).

- As mentioned by the authors in the Discussion, the other connexin Cx43 (Ref 36, Jiang et al 2010) is playing a significant role in HSPC specification in the zebrafish and is expressed in zebrafish arterial cells at 24 hpf. Hence there may be some functional redundancy between Cx43 and Cx48.1, as supported by previous work from the authors showing that a null mutant of Cx48.1 does not exhibit any phenotype in HSPC specification (Ref12, Cacialli et al 2021). This may be problematic for the experiments using drug treatments in the present work, because they are not selective for the different connexins (ex: anti-oxidants (NAC), connexin blockers (heptanol, CBX)), thus blurring interpretations on the specific function of Cx48.1 versus the ones exerted by Cx43 (this should be also valid for the vhl MO treatments). This comment is strengthened by the fact that the authors do not systematically address, for both WT and mutant embryos (Fig3 E, F; FigS6; FigS8), if expression levels with drugs/H2O2/MO are different for the 2 conditions (if relatively equal, it would indeed indicate that these drugs/conditions possibly act on another connexin, which would help the authors in their analyses and interpretations).

****Minor comments****

- The authors should take care of the fact that at 16hpf, it is an overstatement to speak of an aorta when the cord is starting to lumenize at around 18hpf, Jin et al Development 2005 (see Main text referring to Fig3). To make the data more convincing on the ROS production in the ventral side of the cord in wild type embryos (which suggests that future hemogenic cells are already ventralized at that stage), it would be important to obtain confocal images of the region of interest and perform reconstitution of Z-stacks with a sagittal view (rather than longitudinal). It would be nice also to obtain comparable images later on, after lumenization and before initiation of HSPC emergence (before 28hpf).

- The authors may try to rescue the wt phenotype by expressing, in the Cx48.1tq/tq mutants, the mRNA encoding for the wt protein.
- It would be more informative and secure, Fig2D, to show images of the double transgenics (Cx48.1:eGFP;kdrl:mCherry) at 28-30 hpf (rather than 48 hpf) which is more narrowed down to the specification of the hemogenic endothelium thus preventing any risk to visualize the fluorescence signals coming from recently born HSPCs rather than signals from cells embedded in the aortic floor.

Significance

Petzold et al propose a potentially appealing function of connexin Cx48.1 expressed in the zebrafish in the specification of the vascular aortic subtype of cells that will ultimately lead to the formation of hematopoietic stem cell precursors, ie the hemogenic endothelium. They build the work on a possible translation of the function of connexin Cx40 in mammals that is described to localize to mitochondrial membranes in endothelial cells and promote the production of ROS in mitochondria. They propose a function of mitochondria-derived ROS in stabilizing the Hif1/2-Notch1 pathway that is essential for HSPC precursor specification and that may be extended to developmental hematopoiesis in mammals (the putative ortholog of zebrafish Cx48.1 in mammals (Cx40) is highly expressed in the hemogenic endothelium of mouse and human species (see the Discussion paragraph)).

The proposed model is potentially of high significance for the field of hematopoiesis and more generally for translation of knowledge to regenerative medicine aimed at producing hematopoietic stem cells endowed with long term regenerative potential. However, the current work remains preliminary, suffering from lack of resolution in the main experimental axes that are undertaken (WISH analyses and their low accuracy quantifications; low resolution of in situ live imaging; apparent weaknesses of methodologies that are difficult to fully appreciate since poorly detailed in the Method section, in particular regarding WISH quantification and, hence, statistical significance). My recommendation is that the authors should put some efforts in completing the work with other, more quantitative, methodologies (ex: qRT-PCR) and improving the quality/resolution of imaging (by providing confocal images to alleviate any ambiguity on what is visualized and strengthen the results); these are technical approaches that are relatively standard in the field and the authors have extensively used qRT-PCR and FACS-sorting in their previously published work. Also, the endogenous expression of Cx48.1 in the hemogenic endothelium, during the time-window of its specification (20-28hpf), should be addressed; this would be essential to complement the imaging performed with the new transgenic line that expresses eGFP under the control of the Cx48.1 promoter and which provides weak fluorescence signals).

Reviewer 3:

Evidence, reproducibility and clarity

The authors have successfully shown how disruption in connexin (cx)41.8 results in delayed gata2b expression due to Hif1/2a instability in the absence of mitochondrial ROS. The data is presented well, and the paper is written clearly. The paper is well structured, and the data supports the authors' argument. This study provides a valuable contribution to the field.

Could the authors clarify the following questions:

1. In the Results section that describes the delay in gata2b expression (page 4 and Supp. Fig. 4), the authors show that the mutant embryos start expressing more gata2b at 30 - 36hpf after the decreased expression at earlier time points, with no difference at 48hpf. What could explain that recovery? The authors showed that gata2b expression can be rescued by ROS induction in the dose-dependent manner (page 6 and Fig.3 and Supp. Fig. 6). Is this what rescues gata2b expression at 30hpf in the cx41.8 mutants?
As Gata2 has been shown to be a positive autoregulator of itself in mice (Nozawa 2009, Katsumura 2016) and might do so in zebrafish (Dobrzycki 2020), so could gata2b recover itself, in a dose-dependent manner, without the Hif-Nothc1 axis once enough of it is expressed?
2. Does MO-mediated knockdown of vhl in the wildtype and mutant (page 7and Fig. 4) result

in more HSPCs, following the increase in *gata2b* expression from WT baseline? Does that high expression persist, or does it drop?

3. Is *Hif1/2a* expression affected in the mutant? Is it expressed normally but then degraded faster due to the absence of mitochondrial ROS or is it less *Hif1/2a* expressed overall? Are any vascular defects in the mutant embryos?

Significance

His study provides a valuable contribution to the field of developmental hematopoiesis.

Original submission

First decision letter

MS Title: Connexin 41.8 governs timely haematopoietic stem and progenitor cell specification

Authors: Tim Petzold, Sarah, Brivio, Tanja, Linnerz, Masakatsu Watanabe, Holger Gerhardt, Julien Y. Bertrand

Thanks for sending your manuscript via Review commons for consideration. I have now read the manuscript, the reviews and your revision plan and consider that the study is potentially suitable for publication if the reviewers are satisfied with the revisions you make. In your revision plan, you suggest addressing some, but not all, of the reviewer suggestions, some of which you consider would involve experiments that would take too long to undertake. This is reasonable though of course the reviewers will still need to be convinced of the robustness of your data and the conclusions you draw from this data.

Please attend to all of the reviewers' comments and ensure that you clearly highlight all changes made in the revised manuscript. Please avoid using 'Tracked changes' in Word files as these are lost in PDF conversion. I should be grateful if you would also provide a point-by-point response detailing how you have dealt with the points raised by the reviewers in the 'Response to Reviewers' box. If you do not agree with any of their criticisms or suggestions please explain clearly why this is so.

Author responses to Reviewer comments

Manuscript number: RC-2023-02158

Corresponding author: Julien, Y. Bertrand

Please find changes to the manuscript text in the new version highlighted in yellow throughout.

Figure numbers mentioned in responses (red text) refer to those in the new version of the manuscript.

1. General Statements [optional]

Reviewer #1 (Evidence, reproducibility and clarity (Required)):

Summary

The manuscript by Petzold et al. explores the functions of connexin 41.8 (*cx41.8*) (mammalian homologue Connexin 40) in hematopoietic stem cell (HSC) formation in the zebrafish dorsal aorta. The authors use a *cx41.8* allele that appears to be hypomorphic, as the phenotype is milder than a previous *cx41.8* allele that the same group published (Cacialli et al., 2021). *cx41.8^{tq/tq}* mutants exhibit delayed onset of hemogenic endothelial specification, as marked by *gata2b* at 24 hpf, but

HSPC development proceeds normally from 48 hpf onwards. A new reporter line for cx41.8, Tg(cx41.8:GFP), was generated and is expressed in the floor of the dorsal aorta, consistent with the location of hemogenic endothelial cells. Lower ROS production in the whole cell and in the mitochondria was reported in the cx41.8^{tq/tq} mutants, and treatment with ROS enhancers, H₂O₂ and menadione, appeared to rescue the mutant phenotype of reduced HSPCs at 28 hpf. Finally, the authors tested a link between cx41.8 and Hif1 α by pharmaceutically (DMOG/CoCl₂) or genetically (vhl morpholino) inhibiting Hif inhibitors, and observed a rescue of HSPC formation in cx41.8 mutants.

Major comments

- I think it would be important for the authors to address the mechanisms of why cx41.8^{tq/tq} and the other cx41.8^{-/-} (leot1/t1) mutant phenotypes are different, with the latter allele showing more severe phenotypes of increased HSPC apoptosis and reduced HSPCs during later development. The authors speculate the cx41.8^{tq/tq} allele encodes a missense mutation in one of the channel domains, and as such, might be a hypomorph. The authors cited the original paper by Watanabe et al. (2006); however, this paper actually noted that the cx41.8^{tq/tq} allele is likely to be a dominant negative - and as such, should have exhibited a stronger phenotype than the leot1/t1 mutant allele. From the paper: "leotw28 and leotq270 heterozygotes have phenotypes different from that of WT; thus, they represent dominant-negative alleles."

Importantly, no data are shown to provide evidence that the allele is a hypomorph - at minimum, qPCR data should be provided to show whether there is NMD of the mRNA in cx41.8^{tq/tq} mutants.

We would like to thank the reviewer for this comment and suggestion. As the reviewer has rightly pointed out, the cx41.8^{tq/tq} mutation is thought to result in a protein with dominant-negative function (Watanabe et al, EMBO Rep, 2006; Watanabe et al, J Biol Chem, 2016).

In fact, we agree that the mutant cx41.8^{tq/tq} protein acts as a dominant-negative and although the reviewer is right to point out that the cx41.8^{t1/t1} mutant may thus exhibit a stronger phenotype which we found not to be the case (*runx1* expression was found to be normal in the cx41.8^{t1/t1} mutant, Cacialli et al, Nature Communications, 2021), we provided our explanation for this in the discussion of the manuscript:

"The partial functionality of the Cx41.8 channel in cx41.8^{tq/tq} mutants [14] may explain why the HSPC program is eventually induced. However, this could also result from functional redundancy between Cx41.8 and other connexins such as Cx43 or Cx45.6 in the mitochondria, since they are also expressed in zebrafish arterial ECs at 24 hpf [18] and cx43 knockdown has previously been shown to result in an HSPC specification defect in zebrafish [36]. This potential functional redundancy may also provide an explanation as to why HSPCs are specified normally, without any delay, in cx41.8^{t1/t1} embryos [12]. In these null mutants, cx41.8 expression is completely absent but may be functionally compensated by other connexins, whereas in cx41.8^{tq/tq} mutants, although cx41.8 is expressed, its channel function is reduced [14]. Moreover, as Cx41.8 may form heterotypic channels with Cx43 and/or Cx45.6 (and potentially also with others), the function of these chimeric channels would also be altered"

We believe this addresses the reviewers concern regarding this, especially given the fact that Cx43 and Cx45.6 have been found to be expressed in arterial ECs at 24 hpf, as cited in the manuscript. With regards to the reviewer's question about whether there is NMD of the cx41.8 transcript, given that the cx41.8^{tq/tq} mutation is missense and does not result in a premature stop codon (usually required for NMD to be induced, Kurosaki et al, J Cell Sci, 2016), we do not believe that there is NMD of the cx41.8 transcript in cx41.8^{tq/tq} mutants. We have however verified this by carrying out the experiment suggested by this reviewer, qPCR analysis of cx41.8 expression in cx41.8^{tq/tq} embryos and wild-type controls at 48 hpf. The data is shown below. Interestingly, we see higher expression in the cx41.8^{tq/tq} embryos, although this difference is not significant. We do not however believe that it is necessary to include this data in the revised manuscript, for the reasons outlined above, and hence have not included it.

- One major missing component is experimental data that distinguish the gap junction/plasma membrane-related and the mitochondrial membrane-related functions of Cx41.8. This is critical, as the role of Connexins in the mitochondria remains poorly understood (and Connexin 43 is the best understood one). Thus, it is a big claim by the authors that Cx41.8 primarily acts through the mitochondria and not the gap junctions.

Suggested experiment: The authors should generate a fluorophore-tagged Cx41.8 - under a ubiquitous (ubb or actin) or HSPC-/hemogenic endothelium-specific (*gata2b*) promoter to monitor the protein localization of Cx41.8. Providing data on whether Cx41.8 protein indeed localizes to the mitochondria is important to support their claim.

We thank the reviewer for this suggestion, which we agree would be a nice experimental approach to try to investigate whether Cx41.8 does indeed localise to the mitochondria in zebrafish endothelial cells.

However, EGFP fused full-length *cx41.8* has previously been generated and was reported to be nonfunctional, and it was suggested that the amount of localised Cx41.8 is also too small to detect using this approach (Watanabe et al, Pigment Cell Melanoma Res, 2012; Usui et al, BBA Advances, 2021). An EGFP tagged CT-truncated Cx41.8 construct has also been generated and shown to rescue the *cx41.8^{t1/t1}* mutant (Usui et al, BBA Advances, 2021), but EGFP expression again could not be detected using this construct in zebrafish.

As such, since efforts to carry out such an approach have failed in previous attempts and since it has already been demonstrated that CX40 (orthologous to *cx41.8*) localises to the mitochondria of endothelial cells (Guo et al, Am J Physiol Cell Physiol, 2017), we believe that confirmation of Cx41.8 localisation to the mitochondria *in vivo* in zebrafish endothelial cells will be extremely challenging and is beyond the scope of this particular manuscript.

- Related to the above point, the authors should test whether the gap junction function of Cx41.8 is intact in the *cx41.8^{tq/tq}* mutants by assessing calcium waves in the GCamp transgenic line.

We agree with the reviewer that this would be a very elegant approach in order to analyse whether Cx41.8 channel function is affected in *cx41.8^{tq/tq}* mutants. However, we also feel that this experiment is definitely beyond the scope of this manuscript. We believe there is already strong published evidence that the *cx41.8^{tq/tq}* mutant results in disrupted channel function (Watanabe et al, EMBO Rep, 2006), as already cited in our manuscript. However, since then, we have also now found additional published *in vivo* evidence that *cx41.8^{tq/tq}* channel function is reduced, which is now also cited in the new version of the manuscript.

- The quantification data in this manuscript are not satisfactory. The authors only provide graphs that show embryos with "low", "medium" and "high" numbers of HSPCs, which is incredibly subjective. Considering that the authors already have the *cx41.8^{tq/tq}* in the Tg(myb:GFP) background (Figure 1E), they could have quantified the precise numbers of Tg(myb:GFP)-positive cells at different timepoints and with the different pharmaceutical rescue experiments. Ideally, this should be combined with other HSPC markers such as Tg(cd41:GFP) or Tg(runx1:GFP) - although this could be limited by the authors' access to the lines or time it takes to cross the mutants to the transgenes.

We thank reviewer 1 for their concern regarding this. We did not have the personnel or resources to perform all of the suggested experiments, including the pharmacological rescue experiments. However, since we already had the *cx41.8^{tq/tq} cmyb:GFP* zebrafish line, we performed the same experiment as performed previously to generate figure 1E (now Supplementary Figure 3A) at a number of different timepoints, including earlier timepoints. The *cmyb:EGFP* transgene marks nascent HSPCs from 28 hpf, and so we quantified differences in budding HSPCs in *cx41.8^{tq/tq} cmyb:EGFP* and *cmyb:EGFP* controls between 28 hpf and 50 hpf (Figure 1E). This data shows that whilst there is initially a reduced number of *cmyb:EGFP⁺* budding HSPCs (between 28 and 30 hpf) in *cx41.8^{tq/tq}* embryos, there is actually an increase in *cmyb:EGFP⁺* budding HSPCs in *cx41.8^{tq/tq}* embryos at 48 hpf and 50 hpf. This corroborates the *in situ* hybridisation data (supplementary Figures 4 and 5) nicely and the slight differences in timings between the *in situ* hybridisation data and that when using the *cmyb:EGFP⁺* line can be explained by the difference in the marker investigated (*gata2b* is expressed prior to *cmyb* during HSPC development) and when using the *cmyb:EGFP⁺* line, a delay in expression of the reporter is also to be expected since it takes some time for EGFP to fold.

- The link between *cx41.8* and Hif1 α is tenuous. The authors should perform *in situ* hybridization for the *hif1* genes and their downstream effector *notch1* which is known to be important for the HSPC specification (Gerri et al., 2018).

We thank the reviewer for this point. we do not expect *hif1/2a* expression to be affected in this mutant. Mitochondrial ROS has been shown to stabilise Hif1/2 α at the protein level, not the mRNA level. Our data, and that of others (Harris et al, Blood, 2013), suggest that in the absence of mitochondrial ROS, prolyl hydroxylases are not inhibited by mitochondrial ROS, and they target Hif1/2 α for ubiquitination and subsequent destruction in a Vhl-dependent manner. We have changed the text in the manuscript to clarify that Hif is stabilised on the protein level (please see the section below). To verify this, we carried out qPCR analysis for *hif2aa* and found its expression to be unaltered between *cx41.8^{tq/tq}* mutants and control embryos (Supplementary Figure 10A).

Since we do however expect *notch1* gene expression to be altered in our mutant embryos, as they are transcriptionally regulated by Hif1/2 α (Gerri, Blood, 2018), we also performed qPCR analysis of *notch1b* between 24 and 48 hpf in *cx41.8^{tq/tq}* mutants and controls to clarify this point and solidify our model (Supplementary Figure 10B). We found that at 24 and 36 hpf there is reduced *notch1b* expression in the *cx41.8^{tq/tq}* mutants relative to control embryos, whilst the expression of *notch1b* increases at 48 hpf. We hypothesise that at 24 and 36 hpf the difference is not statistically significant due the high variation in expression between replicates. As such, this data fits with our model that *notch* expression is switched on later during development in *cx41.8^{tq/tq}* mutants relative to controls, since Hif1/2 α protein stabilisation occurs at a later developmental timepoint in these embryos.

The authors might also want to consider performing transcriptomic analysis (bulk RNA sequencing) from purified HSCs in wild types and *cx41.8* mutants and assess the downstream pathways affected by the loss of this gene.

Although this is an interesting proposition, we consider this suggestion to be out of the scope of this manuscript, especially since our model involves changes in gene expression upstream of HSPC induction. However, we decided to investigate some of the key genes thought to be affected in *cx41.8^{tq/tq}* mutants, including *notch1b*, *gata2b*, *runx1* and *cmyb* by qPCR as discussed above and have included the relevant data in the new version of the manuscript.

Reviewer #1 (Significance (Required)):

Overall, this study presents another piece of evidence that Connexin 41.8 regulates HSC formation. It provides a potential link between Connexin 41.8, mitochondrial ROS regulation and Hif/hypoxia-sensitive pathways in promoting endothelial-to-hematopoietic transition. The role of the mitochondrial ROS in particular is quite interesting and might provide a new angle into the role of connexins in regulating hemato-vascular development; however, the authors would need to strengthen the link between Cx41.8 and mitochondrial respiration.

It is important to note that the quantitative data in this manuscript need to be strengthened and refined to strengthen the conclusions. The study is not very deeply mechanistic and appears to be more at an observational/correlational level. The manuscript might be of interest for people in the hematopoietic field but does not shed much more insight into the cellular and molecular mechanisms that govern HSC formation, particularly in light of the paper on Cx41.8 role by the same group (Cacialli et al., 2021).

Reviewer #2 (Evidence, reproducibility and clarity (Required)):

Summary

Petzold et al are here addressing the potential function of the connexin Cx48.1, a protein involved in the structure of gap junctions, in the specification of future hematopoietic stem cells and progenitors (HSPCs). This piece of work is complementing their previous results showing the function of this connexin isoform in HSPC expansion in the transient hematopoietic niche in the caudal tissue of the zebrafish embryo. They explore phenotypes triggered by the expression of a mutant form bearing a single amino-acid substitution in the fourth transmembrane domain of the protein. Using whole mount in situ hybridization (WISH) of the two transcription factors Gata2b and Runx1, a novel transgenic fish line that expresses eGFP under the control of the Cx48.1 promoter region, and a series of drug treatments interfering with, or promoting, the formation of reactive oxygen species (ROS) production and oxidative stress, they propose that Cx48.1 is also involved upstream of HSPC amplification, rather in their specification at the level of the hemogenic endothelium constituting the ventral floor of the dorsal aorta. Mechanistically, they hypothesize that this function relies on mitochondria-derived ROS that would destabilize the VHL protein involved in mediating the degradation of Hif1/2a transcription factors, thereby stabilizing the Hif1/2a-Notch1a/b signaling axis involved in specification of the hemogenic endothelium.

Major comments

My major comments on the work are on the accuracy of the data in regard to the two main experimental approaches used by the authors and their subsequent analysis/quantification.

1- the WISH and quantitative analyses.

Most of the quantitative analyses in the work are based on chromogenic WISH, which is not sufficiently accurate because leading to highly variable results, in addition to its lack of linearity. WISH is also subjected to important variations, particularly for transcription factors that are expressed at low levels such as Runx1, and to some extent Gata2b also. One obvious example in the paper is the inconsistency of signals that are observed Fig1C (Gata2b, left, wt, 24hpf) and Fig53B (Gata2b, left, wt, 24hpf) in which the signal is barely visible and is comparable to the signal for the cx41.8tq/tq mutant Fig1C, right.

In addition, in the timings that are analyzed in Fig53 (Gata2b, 26 and 28hpf) to argue on temporal delay of expression in the cx41.8tq/tq mutant, the Gata2b signal is masked by the strong increase in tissues other than the hemogenic endothelium in the dorsal aorta (including signal in the somites as well as, possibly, increase in background). In this very example, it is legitimate to question the accuracy of the quantification methodology when the signal in the tissue of interest is drowned in the overall signal from surrounding tissues; how can the authors explain the 100% of embryos that have a 'Low' signal in the region of interest (Fig53C, cx41.8tq/tq mutant in comparison to WT)? This point is also valid for the data quantified Fig54 in which the fitting between WISH data and the quantifications appears to be questionable (for all timing points: 30, 32, 36, 48hpf and comparing mutant with WT).

Importantly, it appears also that all over the WISH quantifications, the reader cannot appreciate the accuracy of the categories High/Medium/Low, which is not at all developed in the Methods section (paragraph Image processing and WISH phenotypic analyses); hence, it is not possible to evaluate the accuracy/validity of statistics in particular in the experiments in which the quantification into these categories is used for CoCl₂ and morpholino analyses to address the contribution of the Hif1/2a-Notch1a/b pathway Fig4 (these experiments generating results that are not as 'black and white' than the other ones in the paper, hence requiring more accuracy; for example, are the differences in the quantification (% of embryos) significant between the WT+vhl MO and Cx41.8^{tq/tq} mutant + vhl MO? Comparing the 2 WISH results for those conditions does not appear to be very convincing).

We have developed the Methods section (paragraph Image processing and WISH phenotypic analyses), which was highlighted as a concern by this reviewer, in order to detail exactly how we performed our image analysis and statistical analyses using this approach. We believe this will satisfy the concerns reviewer 2 has regarding this, we and appreciate that they have a point that this was indeed underdeveloped in the original submission.

My suggestion would be to complement the WISH data and improve the quantitative analyses using another, more accurate approach such as qRT-PCR for example (on dissected trunk regions and, if necessary because of expression in other surrounding tissues (in the case of Gata2b at later time points), after FACS-sorting using a fish line expressing a fluorescent reporter driven by a vascular promoter, ex: the kdrl:mCherry line used in the work). This is particularly important for the expression of the two transcription factors Runx1 and the more upstream Gata2b, the latter being involved in HSPC specification which is taken as a reference. qRT-PCR experiments should be feasible relatively easily and in a reasonable time frame as the technics is not very time consuming and easily accessible.

We thank reviewer 2 for their concerns regarding the *in situ* quantifications used during this study. Although the approach we have used is widely used in the field to quantify gene expression differences, we appreciate that our data could be strengthened by complementing it with another approach. As such we have done the following:

We have complemented our *in situ* hybridisation characterisation of delayed hemogenic endothelium formation and HSPC specification with qPCR experiments (Supplementary Figure 6). For this, we dissected the trunks and tails of *cx41.8^{tq/tq}* embryos and controls and performed qPCR analysis of *gata2b*, *runx1* and *cmyb* expression at different timepoints during development. This newly added qPCR data for *gata2b*, *runx1* and *cmyb* expression in *cx41.8^{tq/tq}* mutant and control trunk and tails corroborates the *in situ* hybridisation data excellently for nearly all the timepoints tested and so we thank the reviewer for their suggestion and agree that the newly added qPCR data have added robustness to our findings.

Finally, there is a confusion in the quantification regarding the number of HSPCs (see the beginning of the second paragraph of Results 'The HSPC specification defect in *cx41.8^{tq/tq}* mutants is due to a delay in Gata2b expression') and the % of embryos falling into the 3 categories High/Medium/Low FigS2, *cmyb* 48hpf. The authors use this argument (based on the WISH *cmyb* signals) to infer that the deficit in the *cx41.8^{tq/tq}* mutant is not due to controlling HSPC number (no difference in *cmyb* between WT and mutant) but rather upstream, at the level of the hemogenic endothelium, which is not a thorough argument at that point.

We thank reviewer 2 for pointing this out to us and agree that the wording we used is a little confusing. We have therefore added to the first sentence of the second paragraph in the results section "The HSPC specification defect in *cx41.8^{tq/tq}* mutants is due to a delay in *gata2b* expression" which now reads:

"Hence, since HSPC specification is initially reduced, but then recovers in *cx41.8^{tq/tq}* embryos, we suspected a delay in the formation of the haemogenic endothelium in these mutants. To test this hypothesis..."

We believe this change to the manuscript will satisfy the reviewers concern by making this section more logical for the reader.

2 - Fluorescence imaging and associated interpretation/conclusions.

The fluorescence images (Fig1E; Fig2B,D; Fig3A) are very difficult to analyze; they lack resolution because they appear to be epifluorescence images and not confocal images. When the signal is low, which is in particular the case for the novel *Cx41.8:EGFP* fish line, Fig2B (which is confirmed with the FACS GFP signal in comparison to the mCherry of the *kdrl:mCherry* fish line), it is not possible to provide convincing images on the vascular/aortic expression because of the high background of diffusion (the authors state 'likely to be the aortic floor', indeed it is not possible to validate the fact that the expression is truly in potential hemogenic cells). The double positive population in the FACS (Fig2C, right) does not resolve the issue because if indeed *cx41.8* is expressed in endothelial cells (as expected from previous studies), the double positive population could equally be endothelial cells from inter-somitic vessels, for example (not to mention the underlying vein which is very close to the aorta in the trunk)). Fig2D, images are too small and, again, the resolution is not good enough to say that double positive cells are on the aortic floor. It is recommended to convince the reader that the authors try to confirm their statements by using confocal microscopy and increase the magnification of the relevant regions of interest.

We thank this reviewer for this point. We have addressed this concern by using, as they suggest, confocal microscopy to try to get higher resolution images.

We have used confocal microscopy to image the *cx41.8:EGFP* line as was done in the previous version of the manuscript, in order to obtain higher resolution images of expression of *cx41.8* in the floor of the aorta at different developmental timepoints - 24, 28 and 48 hpf (Figure 2B). There is an enrichment in *cx41.8:EGFP* expression in flat cells resembling endothelial cells in the floor of the dorsal aorta at 24 hpf (prior to HSPC specification), and at 28 hpf (at the onset of HSPC specification). At 48 hpf (during HSPC budding) *cx41.8:EGFP* expression is present in rounded cells at the floor of the dorsal aorta resembling budding HSPCs.

We have also performed confocal microscopy to image the *cx41.8:EGFP;kdrl:mCherry* line, in order to gain higher resolution images. We did this at 2 different developmental timepoints, 24 and 28 hpf. We have included these images in the new version of the manuscript (Figure 2D). Although we also used confocal imaging for the 48 hpf timepoint, we have kept the same image in the new version that was present for the 48 hpf timepoint from the previous manuscript version since we concluded that it was the highest quality image from all that we have. We have increased the region of interest in this image as suggested by this reviewer.

There is an inconsistency in the data between Fig1E (40hpf, in vivo imaging using the *cmyb:GFP* fish line) and Fig52 (48hpf, WISH *cmyb*); how can we observe 'HSPCs budding from the dorsal aorta' (see legend Fig1, arrowheads) which seems very much decreased in the imaging experiment for the *cx41.8^{tq/tq}* mutant in comparison to WT, and have no effect on the *cmyb* signals Fig52B? What are the GFP⁺ cells that are aligned along the elongated yolk Fig1E and that appeared to be decreased in number in the mutant?

We agree that this disparity is confusing for the reader. We believe the disparity between these results is due firstly to the fact that the experiment in Supp. Fig 2B was performed 8 hours after that in Supplementary Fig 3A, and secondly due to the time it takes for GFP to fold (in the case of Supplementary Fig 3A). It is also important to keep in mind that the phenotype is not a complete absence of HSPC budding, but only a delay in the onset of EHT.

We have however addressed this concern by carrying out the experiment described above - we have performed the same experiment as performed previously to generate Supplementary Fig 3A but also at earlier timepoints. The *cmyb:EGFP* transgene marks nascent HSPCs from 28 hpf, and so we have quantified differences in budding HSPCs in *cx41.8^{tq/tq} cmyb:EGFP* and *cmyb:EGFP* controls at numerous timepoints - 28, 30, 32, 48 and 50 hpf (Figure 1E).

Regarding the GFP⁺ cells aligned along the yolk in Supplementary Figure 3E, we thank the reviewer for pointing this out. These cells are multiciliated cells (MCCs), from the kidney tubules (Wang et al, Development 2013). We decided to quantify the numbers of these multiciliated cells in order to determine whether there is indeed a difference in their numbers between *cx41.8^{tq/tq};cmyb:EGFP*

and *cmyb:EGFP* control embryos. We found that there is indeed a reduction in the number of multiciliated cells in *cx41.8^{tg/tg};cmyb:EGFP* embryos relative to controls at 28, 30, 32, 48 and 50 hpf). We have included these findings in the supplementary data (Supplementary Figure 3B) of the new manuscript and have also mentioned them in the relevant section of the text since it may be of interest to the community working on multiciliated cells, but since we do not believe this data to be of further relevance for the remainder of the findings in our manuscript, it is not discussed further.

Are the authors sure of their statement on budding HSPCs when the GFP signal pointed by arrows could in majority be hemogenic cells? (which would be in favor of their hypothesis on Cx41.8 being involved rather in hemogenic endothelium/HSPC specification).

Since *cmyb* is a marker of HSPCs and not of the haemogenic endothelium as demonstrated in numerous publications (North et al, Nature, 2007; Bertrand et al, Development, 2008; Bertrand et al, Nature, 2010 and others), we are confident that this transgene is marking nascent HSPCs and not the hemogenic endothelium.

Other Major Comments:

- It would be important to investigate/show, at least with qualitative WISH experiments all along the time-window of HSPC specification as stated by the authors (26-54hpf, see main text third paragraph of Results), that Cx41.8 is detected in arterial endothelial cells (and perhaps enriched in the hemogenic endothelium?), in complement to the work they are referring to on transcriptomic data at 24hpf (Ref18 Gurung et al Sci Rep 2022). Ideally, these WISH data should be resolute enough to provide clear localization in aortic cells versus cells in the aortic floor to bring significant added value to the work that lacks spatial resolution (ex: fluorescent WISH using confocal microscopy, allowing to superpose signal with cell types (either by double fluorescent WISH (vascular marker + Cx41.8) or superposing fluorescence signals with transmitted light)).

We agree with this reviewer regarding this point. As also stated above, we have used confocal microscopy to image the *cx41.8:EGFP* line as was done previously (Fig 2B), in order to obtain higher resolution images of expression of *cx41.8* in the floor of the aorta at different developmental timepoints (24, 28 and 48 hpf, see Figure 2B). There is an enrichment in *cx41.8:EGFP* expression in flat cells resembling endothelial cells in the floor of the dorsal aorta at 24 hpf (prior to HSPC specification), and at 28 hpf (at the onset of HSPC specification). At 48 hpf (during HSPC budding) *cx41.8:EGFP* expression is present in rounded cells at the floor of the dorsal aorta resembling budding HSPCs (Figure 2B).

We have also performed confocal microscopy to image the *cx41.8:EGFP;kdrl:mCherry* line as was done for in Fig 2D of the previous version, in order to gain higher resolution images. We did this at 2 different developmental timepoints, 24 and 28 hpf. We have included these in the new version of the manuscript (Fig 2D), but in our opinion, the image that was present for the 48 hpf timepoint in Fig 2D previously is still the best overview image and so we have retained it in the main figures of the manuscript. We have increased the region of interest in this image as suggested by this reviewer.

We believe that this confocal microscopy imaging is sufficient to robustly demonstrate, along with the FACS data in the manuscript and the scRNA-seq data that we have cited (Ref19 Gurung et al Sci Rep 2022), that *cx41.8:EGFP* is expressed in arterial endothelial cells during the timeframe of HSPC specification and that its expression is enriched in the floor of the aorta.

- As mentioned by the authors in the Discussion, the other connexin Cx43 (Ref 36, Jiang et al 2010) is playing a significant role in HSPC specification in the zebrafish and is expressed in zebrafish arterial cells at 24 hpf. Hence there may be some functional redundancy between Cx43 and Cx48.1, as supported by previous work from the authors showing that a null mutant of Cx48.1 does not exhibit any phenotype in HSPC specification (Ref12, Cacialli et al 2021). This may be problematic for the experiments using drug treatments in the present work, because they are not selective for the different connexins (ex: anti-oxidants (NAC), connexin blockers (heptanol, CBX)), thus blurring interpretations on the specific function of Cx48.1 versus the ones exerted by Cx43 (this should be also valid for the vhl MO treatments). This comment is strengthened by the fact that the authors do

not systematically address, for both WT and mutant embryos (Fig3 E, F; FigS6; FigS8), if expression levels with drugs/H₂O₂/MO are different for the 2 conditions (if relatively equal, it would indeed indicate that these drugs/conditions possibly act on another connexin, which would help the authors in their analyses and interpretations).

We thank the reviewer for these comments and we agree with their concerns regarding the possibility of other Connexins being affected by our experiments using drug treatments. However, we do not rule this out in our manuscript and actually discuss it as being a very realistic prospect, as written about in the discussion section.

Sadly, to the best of our knowledge, no selective Cx41.8 inhibitors have been described for use in zebrafish, otherwise we would of course have used these. Hence, this was the reason for our choice of compounds, many of which we also used in our previous publication, although at different timings/stages (Cacialli et al, Nature Communications, 2021).

The haemogenic endothelium/HSPC phenotype in *cx41.8^{tq/tq}* embryos confirms that this connexin plays a role in HSPC specification, whilst we believe disentangling which other connexins are also involved in this process will be interesting to look into in other future studies but is beyond the scope of this one - we believe that together, the data presented in our manuscript, along with the revisions that we have carried out, are convincing to demonstrate a role for Cx41.8 in the mechanism that we describe.

Minor comments

- The authors should take care of the fact that at 16hpf, it is an overstatement to speak of an aorta when the cord is starting to lumenize at around 18hpf, Jin et al Development 2005 (see Main text referring to Fig3).

We thank the reviewer for this clarification. We have changed the relevant text to state “vascular cord” instead of “aorta” and have mentioned that it begins to lumenize around 18 hpf for clarification. We have also added the suggested reference.

To make the data more convincing on the ROS production in the ventral side of the cord in wild type embryos (which suggests that future hemogenic cells are already ventralized at that stage), it would be important to obtain confocal images of the region of interest and perform reconstitution of Z-stacks with a sagittal view (rather than longitudinal). It would be nice also to obtain comparable images later on, after lumenization and before initiation of HSPC emergence (before 28hpf).

We thank the reviewer for this suggestion. Although we agree that the suggested experiments would further solidify our data, we believe they are beyond the scope of the present manuscript. We believe that the data provided in Figures 3 B and C of the new manuscript are sufficient to show that both ROS and mitochondrially derived ROS are present in endothelial cells of the vascular cord at 16 hpf, prior to the induction of HSPC specification and that a more detailed analysis with numerous developmental timepoints would be interesting for additional future studies.

- The authors may try to rescue the wt phenotype by expressing, in the Cx48.1tq/tq mutants, the mRNA encoding for the wt protein.

Although we appreciate this suggestion, we do not believe that this experiment would add much in terms of value to the conclusions of our manuscript and, as such, we believe this suggestion is surplus to requirements for this manuscript.

- It would be more informative and secure, Fig2D, to show images of the double transgenics (Cx48.1:eGFP;kdr1:mCherry) at 28-30 hpf (rather than 48 hpf) which is more narrowed down to the specification of the hemogenic endothelium thus preventing any risk to visualize the fluorescence signals coming from recently born HSPCs rather than signals from cells embedded in the aortic floor.

We thank the reviewer for this suggestion and agree that this is a useful addition to the manuscript. We have performed the experiments that they have suggested. As also discussed above, we have performed confocal microscopy to image the *cx41.8:EGFP;kdrl:mCherry* line as was done for in Fig 2D of the previous version, in order to gain higher resolution images. We did this at 2 additional developmental timepoints, 24 and 28 hpf. We have included these in the new version of the manuscript (Fig 2D).

Reviewer #2 (Significance (Required)):

Significance

Petzold et al propose a potentially appealing function of connexin Cx48.1 expressed in the zebrafish in the specification of the vascular aortic subtype of cells that will ultimately lead to the formation of hematopoietic stem cell precursors, ie the hemogenic endothelium. They build the work on a possible translation of the function of connexin Cx40 in mammals that is described to localize to mitochondrial membranes in endothelial cells and promote the production of ROS in mitochondria. They propose a function of mitochondria-derived ROS in stabilizing the Hif1/2-Notch1 pathway that is essential for HSPC precursor specification and that may be extended to developmental hematopoiesis in mammals (the putative ortholog of zebrafish Cx48.1 in mammals (Cx40) is highly expressed in the hemogenic endothelium of mouse and human species (see the Discussion paragraph)).

The proposed model is potentially of high significance for the field of hematopoiesis and more generally for translation of knowledge to regenerative medicine aimed at producing hematopoietic stem cells endowed with long term regenerative potential. However, the current work remains preliminary, suffering from lack of resolution in the main experimental axes that are undertaken (WISH analyses and their low accuracy quantifications; low resolution of in situ live imaging; apparent weaknesses of methodologies that are difficult to fully appreciate since poorly detailed in the Method section, in particular regarding WISH quantification and, hence, statistical significance). My recommendation is that the authors should put some efforts in completing the work with other, more quantitative, methodologies (ex: qRT-PCR) and improving the quality/resolution of imaging (by providing confocal images to alleviate any ambiguity on what is visualized and strengthen the results); these are technical approaches that are relatively standard in the field and the authors have extensively used qRT-PCR and FACS-sorting in their previously published work. Also, the endogenous expression of Cx48.1 in the hemogenic endothelium, during the time-window of its specification (20-28hpf), should be addressed; this would be essential to complement the imaging performed with the new transgenic line that expresses eGFP under the control of the Cx48.1 promoter and which provides weak fluorescence signals).

Reviewer #3 (Evidence, reproducibility and clarity (Required)):

The authors have successfully shown how disruption in connexin (*cx*)41.8 results in delayed *gata2b* expression due to Hif1/2a instability in the absence of mitochondrial ROS. The data is presented well, and the paper is written clearly. The paper is well structured, and the data supports the authors' argument. This study provides a valuable contribution to the field.

Could the authors clarify the following questions:

1) In the Results section that describes the delay in *gata2b* expression (page 4 and Supp. Fig. 4), the authors show that the mutant embryos start expressing more *gata2b* at 30 - 36hpf after the decreased expression at earlier time points, with no difference at 48hpf. What could explain that recovery?

We thank reviewer 3 for this question. The partial functionality of the Cx41.8 channel in *cx41.8^{tq/tq}* mutants may explain why the HSPC program is eventually induced (leading to sufficient mitochondrial ROS production for Hif1/2 α stabilisation). However, this could also result from functional redundancy between Cx41.8 and other connexins such as Cx43 or Cx45.6 in the mitochondria, since they are also expressed in zebrafish arterial ECs at 24hpf (Gurung et al, Sci Rep, 2022) and *cx43* knockdown has previously been shown to result in an HSPC specification defect in zebrafish (Jiang et al, Fish Physiol Biochem, 2010). Together, these aspects may explain the

recovery, although delayed, of *gata2b* expression in the *cx41.8^{tq/tq}* mutant, as discussed in detail in our manuscript.

The authors showed that *gata2b* expression can be rescued by ROS induction in the dose-dependent manner (page 6 and Fig.3 and Supp. Fig. 6). Is this what rescues *gata2b* expression at 30hpf in the *cx41.8* mutants?

This is exactly right, we hypothesize that in *cx41.8^{tq/tq}* mutants, it takes longer for mitochondrial ROS production to reach above the threshold required to stabilise Hif1/2 α and hence induce *gata2b* expression, which is supported by the data referred to by this reviewer.

As Gata2 has been shown to be a positive autoregulator of itself in mice (Nozawa 2009, Katsumura 2016) and might do so in zebrafish (Dobrzycki 2020), so could *gata2b* recover itself, in a dose-dependent manner, without the Hif-Notch1 axis once enough of it is expressed?

We thank reviewer 3 for this question/suggestion. We believe that our data show that Cx41.8 is required for mitochondrial ROS production, which stabilises Hif1/2 α and switches on downstream *gata2b* via Notch1a/b (which will be added, please see above). As such, we believe that the Hif1/2 α /Notch1a/b axis is required, at least for the initial induction of *gata2b* expression. However, reviewer 3 makes a very interesting point regarding the potential for *gata2b* to positively autoregulate itself, which may of course occur once *gata2b* expression has been induced by the Cx41.8-mitoROS-Hif1/2 α -Notch1a/b-*gata2b* pathway. We thank the reviewer again for this interesting proposition and have added this suggestion into our discussion in the following paragraph:

“GATA2 has been shown to positively autoregulate its own expression in mice (Nozawa et al, Genes to Cells, 2009; Katsumura et al, Cell Reports 2016), and Gata2b may also act in this way in zebrafish (Dobrzycki et al, Commun Biol, 2020). Therefore, one can speculate that once *gata2b* expression has been induced by the Cx-mitoROS-Hif1/2 α -Notch1a/b-*gata2b* pathway, it may also further activate its own expression, increasing robustness of the haematopoietic transcriptional program.”

2) Does MO-mediated knockdown of *vhl* in the wildtype and mutant (page 7 and Fig. 4) result in more HSPCs, following the increase in *gata2b* expression from WT baseline? Does that high expression persist, or does it drop?

This is an interesting question. We had already clarified this in the case of *cx41.8^{tq/tq}*, since we showed that the *vhl* MO results in more HSPCs (as determined by *runx1* expression) at 28 hpf (Fig 4D of new manuscript) but we have now added data for the same marker at the same timepoint for WT embryos (Fig 4D).

Although the *vhl* MO results in an increase in *runx1* signal in WT embryos, since the majority of WT embryos injected with the control MO already have “high” *runx1* WISH signal at 28 hpf, the difference between injected and control MO injected WT embryos is not significant (Figure 4D), as can be expected. This is now explained in the manuscript following the relevant data addition.

3) Is Hif1/2 α expression affected in the mutant? Is it expressed normally but then degraded faster due to the absence of mitochondrial ROS or is it less Hif1/2 α expressed overall?

We thank reviewer 3 for this question, which is similar to a point made by reviewer 1. To clarify, we do not expect *hif1/2a* expression to be affected in this mutant. Mitochondrial ROS has been shown to stabilise Hif1/2 α at the protein level, not the mRNA level. Our data, and that of others (Harris et al, Blood, 2013), suggest that in the absence of mitochondrial ROS, prolyl hydroxylases are not inhibited by mitochondrial ROS, and they target Hif1/2 α for ubiquitination and subsequent destruction in a Vhl dependent manner (as shown in Fig. 4 C and D).

To clarify this in the manuscript, we have added qPCR data of *hif2aa* in WT and *cx41.8^{tq/tq}* embryos at 24, 30 and 36 hpf (Supplementary Figure 10A). These data show that *hif2aa* expression does not differ between the 2 genotypes.

We have also adjusted the text in three places (including in the abstract) to clarify that Hif1/2 α is stabilised at the protein level, as shown below. We believe these changes have made this important point more understandable for the reader:

1. "... Mitochondrial-derived reactive oxygen species (ROS) have been shown to stabilise the hypoxia-inducible factor 1/2 α (Hif1/2 α) proteins, allowing them.."
2. "Recent research has demonstrated that hypoxia and mitochondrial ROS are required for the stabilisation of the transcription factors Hif1/2 α at the protein level"
3. "... as mitochondrial ROS generation may eventually reach the threshold required to sufficiently stabilise the Hif1/2 α proteins for downstream"

Are any vascular defects in the mutant embryos?

Our lab previously reported that *cx41.8^{tg/tg}* embryos have faster ISV growth rate (Denis et al, Front Physiol, 2019). However, we found no evidence of a link between the ISV growth rate increase and the HSPC specification defect in these embryos. Importantly, we show that aorta specification is normal in *cx41.8^{tg/tg}* mutants, as determined by *dll4* expression at 24 (Supp. Fig. 1C) and 28 hpf (Supp. Fig. 1D).

Reviewer #3 (Significance (Required)):

His study provides a valuable contribution to the field of developmental hematopoiesis.

Second decision letter

MS Title: Connexin 41.8 governs timely haematopoietic stem and progenitor cell specification

Authors: Tim Petzold, Sarah, Brivio, Tanja, Linnerz, Masakatsu Watanabe, Holger Gerhardt, Julien Y. Bertrand

Dear Julien,

I have now received all the referees' reports on the above manuscript, and have reached a decision. I am sorry to say that the outcome is not a positive one. The referees' comments are appended below, or you can access them online: please go to.

As you will see, although referee 3 is positive, the other two referees still have some significant concerns about your paper, and are not strongly in favour of publication. Given their opinions, I must therefore, reject your paper. I do realise this is disappointing news, but we receive more papers than we can publish, and we can only accept manuscripts that receive strong support from referees.

I do hope you find the comments of the referees helpful.

Comments from the Reviewers:

Reviewer 1: SUMMARY OF THE ADVANCE MADE IN THIS PAPER AND ITS POTENTIAL SIGNIFICANCE TO THE FIELD

This reviewer appreciates the efforts made by the authors to answer to the several points and concerns. However, the revision as it stands is still lacking precision and several important results remain unconvincing, precluding publication as such.

The topic is interesting and the mitochondrial function of connexins on ROS transport original. However, this paper only translates knowledge from other species without showing that *cx41.8* is localized at mitochondrial membranes, which would be necessary to move the field forward and consider the zebrafish model as competitive with mammalian systems to address the biological

functions of these proteins at the mitochondrial level (and, hence, in specification of hemogenic cells).

SUGGESTIONS TO AUTHORS

1. Main concern with the quality of imaging.

- The authors have made a quite significant effort (ex: new panel Fig. 2B, with the cx48.1:EGFP Tg line in which expression in the aortic floor at 24-28 hpf is now convincing).

- However, images Fig. 3B, C remain unconvincing. Apart from the blurred green signal indicating developing endothelial tissues (kdrl:EGFP), we do not know at which region we are looking at precisely (hence the statements lines 230 and 233 saying we are looking at the ventral side of the vascular cord are not correct). As visualized in Jin et al 2005 (Ref 21 in the Ref list), the ventral region of the cord at 18 hpf does not allow discriminating the vein from the artery. Hence, to speak of addressing the haemogenic tissue here is at risk (including for the HSPC program, as stated in the title of this paragraph). The only option the authors have here is to provide the superposition of high phase contrast transmitted light images (DIC) with high resolution confocal images (or, alternatively, another marker highlighting cells in the surrounding tissues), and in cross sections, at 18 (cord stage), 20 (clear lumenization) and 28 hpf (vein/aorta have clearly separated, each with a lumen) to make sure of the localization of signals (cellROX, mitoSOX) at the ventral side of the cord and of the aortic wall. Providing these images is not beyond the scope of this paper I believe, and certainly not out of reach in the timing of a revision.

To finish on this point and consequently to the upper comments, the text lines 330-331 of the last paragraph of the Discussion 'In summary are important for ROS production in the mitochondria of vascular cord ECs, as early as 16 hpf' is an overstatement.

2. Main concern with WISH and quantitative analyses

- Methodology on WISH quantification. This point was also raised as a Major point by Reviewer 1. The authors have substantiated their methodology in the Method section. One should be reluctant with the idea to make statistics based on 'evaluation' instead of measurement, and make sure of the linearity of the quantified signal with time.

- Because of limits in WISH quantifications, the authors have complemented in situ hybridization data with using the cx48.1tq/tq cmyb:EGFP line (in comparison with cmyb:EGFP) and quantified the number of 'budding HSPCs' (Fig. 1E). Consistently with WISH data, they measure a decrease in the apparent number of budding EGFP+ cells at 28 and 30 hpf. While this is rather convincing, I find the data presented Fig. S6 less convincing (the qPCR experiments on dissected trunks + tails), although the authors write in their rebuttal that they fit 'excellently' with the WISH results. There is a technical issue here (apart from the fact that the statistical test are not significant, hence we may only see a tendency): the qPCR have not been performed on FACS sorted cells, as had been suggested (ex: cmyb:EGFP, cx48.1tq/tq cmyb:EGFP sorted cells). Measuring transcription factors such as runx1 and gata2b is not always straightforward, even more when the cells of interest are in minority which is the case for endothelial cells in comparison to all the other cells in the trunk + tail. This is probably why the data are not significant and the qPCR results so variable.

- There are still inconsistencies with quantifications. For example, why do the quantifications Fig. 1E (with the cmyb:EGFP Tg line) do not fit with the WISH data Fig. S2B (48hpf) in which there is virtually no change in cmyb detection between Wt and mutant? In addition, Fig. S2 shows WISH signals and not cell numbers (line 171 states 'HSPC numbers').

3. Other main comments regarding the impact of the work and missing evidence.

Importantly, regarding the cx48.1 protein per se and its mRNA:

- The authors should put some efforts to show that this connexin is indeed translocated at the mitochondrial membrane in zebrafish endothelial cells. This would be an essential piece of evidence to add to the paper to strengthen its impact (hence not beyond its scope considering the importance of the point; the authors provide an entire Figure (Fig. 5) with a model proposing that it would be indeed the case, although not addressed).

This point was also raised as a Major point by Reviewer 1. While the authors answered to the point by arguing that a EGFP fusion was shown by others to be unstable/non-functional, other

alternatives could be tested. EGFP and other fluorescent reporters are relatively large molecules; an alternative is to insert a small tag (HA, V5 etc ...). Tests can then be easily made by injecting the epitope-tag encoding cx48.1 mRNAs at the one cell stage (followed by performing immunofluorescence at 20-28 hpf, during specification of the HE). These tags may be inserted at the N and/or C termini (taking care of the signal peptide) but also in intra- and extracellular loops (see Fig. 1A).

This is not technically challenging and the authors should take the opportunity of performing these experiments to strengthen their work.

- the authors do not show any in situ of cx48.1 (their former work (Ref 18, Denis et al 2019), if I am correct, do not show in situ but only the significant enrichment of the cx48.1 mRNA by qPCR in sorted kdrl:eGFP cells).

4. The authors should be more careful with their writing; several points are clearly overstated:

- A fluorescent Tg line is indicative that a driver has been activated but not that the fluorescent reporter would match the expression window of the endogenous protein (they not necessarily have the same half-life); ex: line 218-219 'Together, this demonstrate that cx48.1 is expressed in presumptive haemogenic ECs and budding HSPCs in the aortic floor during the time of EHT and HSPC specification'. To formally show that cx48.1 is expressed at specific stages (in particular at later time points since EGFP has a relatively long half-life), one would need in situ hybridization (see point 3 above).

- line 226, at the timing of lumenization (18 hpf), it is too early to speak of haemogenic endothelial cells. The cord has not lumenized yet and the cardinal vein and dorsal aorta have not separated physically (see Jin et al DOI: 10.1242/dev.02087).

Note the error in the name of the first author for this paper in the Ref list line 541 (Jin and not Jinn).

Reviewer 2: SUMMARY OF THE ADVANCE MADE IN THIS PAPER AND ITS POTENTIAL SIGNIFICANCE TO THE FIELD

This will be of interest to the field of developmental haematopoiesis, and provide further information to manipulate redox states to modulate haematopoietic output when generating HSPCs in vitro for therapeutical applications.

SUGGESTIONS TO AUTHORS

Petzold et al characterize the phenotype of a second allele of the cx41.8 gene. They also generate a transgene driven by the cx 41.8 promoter and show it is co-expressed in Kdrl-cherry ECs localized in the ventral wall of the DA. Loss of cx41.8 in this mutant led to a decrease in the onset of gata2b expression apparent up to 36hpf, potentially leading to fewer HE cells being specified. By 48 hpf, the phenotype seemed to have recovered, in contrast with their previous results using a different cx41.8 (Cacciali et al, 2021). This mutant seems to behave more like a hypomorphic allele compared to the previous mutant studied. It is clear that the expression of gata2b is delayed, and that seems to be consistent across experiments. While the link to ROS production had been established by the lab already, the new information relates to the potential effect of increased ROS on Hif1/2a protein stability and further regulation of gata2b via notch signalling and therefore impaired specification of hemogenic endothelium in the mutant. The link between ROS and runx1/gata2b expression is also well established with the inhibitor treatments. Overall, while interesting, I feel that this work is more an extension of their previous publication rather than a significant conceptual advancement.

Major comments

1- I understand this has been previously reviewed and that the authors have endeavoured to address the reviewer's comments, especially the ones related to ISH quantitation and verification using qPCR. While this has improved in this version of the manuscript, some of the data and its interpretation lacks robustness. For example, image quantitation of the in situ data for HE markers (Dobrzycki et al, 2018) is a more robust way to assess expression changes from images and has been widely used in the community. The extra qPCR data generated as a response to reviewers is welcome, but the analyses show no statistically significant difference (e.g. Supp Fig 10) between genotypes - while I agree that high variability may be to blame here, more biological samples per genotype would likely resolve this issue. As it is, the conclusions based on this data lack robustness.

2- A central proposition in this manuscript is that Gata2b is the upstream regulator of runx1 and cmyb in HE. However, recent papers from the Zon (Avagyan et al, 2021) and de Pater labs (Gioachinno et al, 2021), demonstrate that, in two independent gata2b mutants, the initial expression of runx1 and cmyb is unaltered in the time window between 24-30hpf, with the first changes in runx1 or myb expression being visible from 33hpf onwards. In other words, lack of gata2b is not sufficient to impair HE specification or EHT events (Gioachinno et al, 2021), and thus the interpretation that gata2b expression delay is impairing timing of HE specification/HSPC emergence lacks good support.

3- Dobrzycki et al (2020) show that gata2a is required for that early expression of runx1 and gata2b (at 23hpf), but gata2b expression recovers to wt levels by 30hpf - do you know what happens to gata2a expression in cx41.8 mutants (in the interval between the increased ROS production and recovery of gata2b expression) - is gata2a expression affected? Given that they have also shown that notch signalling regulates gata2b and runx1 in parallel independently of gata2a, it is possible that the main effect of increased ROS is in decreasing the expression of notch receptors? Would knocking down gata2a in cx41.8 embryos prevent the recovery?

Minor comments

1. Some of the supplementary figures are referred out of order (e.g. panels in supp fig 2 are referred before supp fig 1) - best to re-organize the figures in order to follow the logic of the text.
2. Results section, lines 144-147 - the authors make a distinction between gata2b as HE marker, runx1 and myb as nascent HSPC markers - given that they are all expressed in HE prior to HSPC emergence (i.e. EHT, starting from ~34hpf, Kissa and Herbomel, 2010), they are all HE/nascent HSPC markers. cmyb is indeed used as an HSPC marker, but for later stages (~36hpf onwards)
3. Fig 1E, panels 1 and 2 - the number of cmyb-GFP cells is described as 'budding HSPCs' at 28 and 30hpf, whereas they are only HE at this stage.
4. Related to that, there seems to be an increased number of cmyb-GFP cell at 48 and 50hpf. Could it be more proliferation, or increased number of EHT events? Nevertheless, from the cmyb in situ in the CHT there is no apparent difference between mutant and wt, which seems counterintuitive - do the authors have an explanation for this? Also, if you count cmyb-GFP cells in the CHT, will there be difference between the genotypes?
5. Fig 3A - do you have quantitation data for the increase in ROS? That would help make this point more robust.
6. The treatment with the ROS inducers is an interesting experiment. Do they have the same effect on wt embryos? Similarly what happens in the wt treated with CoCl₂ or DMOG?

Reviewer 3: SUMMARY OF THE ADVANCE MADE IN THIS PAPER AND ITS POTENTIAL SIGNIFICANCE TO THE FIELD

This manuscript brings new insights into how HSC generation is regulated. Here, authors aimed to study the temporal regulation of endothelial-to-hematopoietic (EHT) transition using the zebrafish model. This process is essential to generate hematopoietic stem cells (HSCs) in vivo. They discovered that the EHT is delayed in connexin mutant embryos. They further proposed a role for ROS and HIF in EHT regulation.

SUGGESTIONS TO AUTHORS

This manuscript brings new insights into how HSC generation is regulated. Here, authors aimed to study the temporal regulation of endothelial-to-hematopoietic (EHT) transition using the zebrafish model. This process is essential to generate hematopoietic stem cells (HSCs) in vivo. They discovered that the EHT is delayed in connexin mutant embryos. They further proposed a role for ROS and HIF in EHT regulation.

The paper was reviewed by 3 Reviewers. All questions and comments raised by these Reviewers are logical. The authors addressed the comments and significantly improved the manuscript. For example, both Reviewers 1 and 2 raised concerns about quantification and these authors properly addressed these concerns providing with new figures and writing the Methods in detail which were missing in the previous submission.

Reviewer 1 asked the authors to prove their major claim that Cx41.8 primary acts through the mitochondria and not the gap function and, here, authors response is not very convincing. I understand the technical problems but it is not because Cx40 localises there in human, that in zebrafish is necessarily the same. This is rather speculative, and thus, authors should carefully

conclude on a 'potential' link between Cx41.8 and Ros/Hif to promote EHT, in line with Reviewer 1 comments and propose future experiments.

Author responses to Reviewer comments

Reviewer 1:

...to speak of addressing the haemogenic tissue here is at risk (including for the HSPC program, as stated in the title of this paragraph).

We have changed the title of this paragraph to:

“Temporal mitochondrial ROS induction of HSPCs requires cx41.8”

...Fig. S2 shows WISH signals and not cell numbers (line 171 states 'HSPC numbers').

We have made the relevant change.

A fluorescent Tg line is indicative that a driver has been activated but not that the fluorescent reporter would match the expression window of the endogenous protein (they not necessarily have the same half-life); ex: line 218-219 'Together, this demonstrate that cx48.1 is expressed in presumptive haemogenic ECs and budding HSPCs in the aortic floor during the time of EHT and HSPC specification'.

We have made the relevant change in the text.

line 226, at the timing of lumenization (18 hpf), it is too early to speak of haemogenic endothelial cells. The cord has not lumenized yet and the cardinal vein and dorsal aorta have not separated physically (see Jin et al DOI: 10.1242/dev.02087). Note the error in the name of the first author for this paper in the Ref list line 541 (Jin and not Jinn).

We have changed the text and also updated the citation with the correct spelling of the name.

Reviewer 2:

A central proposition in this manuscript is that *Gata2b* is the upstream regulator of *runx1* and *cmyb* in HE. However, recent papers from the Zon (Avagyan et al, 2021) and de Pater labs (Gioachinno et al, 2021), demonstrate that, in two independent *gata2b* mutants, the initial expression of *runx1* and *cmyb* is unaltered in the time window between 24-30hpf, with the first changes in *runx1* or *myb* expression being visible from 33hpf onwards. In other words, lack of *gata2b* is not sufficient to impair HE specification or EHT events (Gioachinno et al, 2021), and thus the interpretation that *gata2b* expression delay is impairing timing of HE specification/HSPC emergence lacks good support.

We have added a sentence in the discussion to mention that *gata2a* may also be implemented and that further work is required to establish this. We have also cited 2 relevant studies regarding *gata2a*.

Results section, lines 144-147 - the authors make a distinction between *gata2b* as HE marker, *runx1* and *myb* as nascent HSPC markers - given that they are all expressed in HE prior to HSPC emergence (i.e. EHT, starting from ~34hpf, Kissa and Herbomel, 2010), they are all HE/nascent HSPC markers. *cmyb* is indeed used as an HSPC marker, but for later stages (~36hpf onwards)

We have made the relevant change to the text.

Fig 1E, panels 1 and 2 - the number of *cmyb*-GFP cells is described as 'budding HSPCs' at 28 and 30hpf, whereas they are only HE at this stage.

We have made the relevant change to the text and have also changed Figure 1E (28 and 30 hpf) so that the Y axis is labelled with “cmyb GFP+ HE cells”.

Reviewer 3.

This is rather speculative, and thus, authors should carefully conclude on a 'potential' link between Cx41.8 and Ros/Hif to promote EHT

We have made the relevant change to the text - we have changed provides “strong evidence” to provides “evidence” in the final sentence prior to the discussion in order to avoid potential overstating. However, we do believe that our manuscript provides evidence for this link and so we believe retaining the words “provides evidence” is fair

Transfer to Biology Open

First decision letter

MS ID#: bio.062118

MS Title: Connexin 41.8 governs timely haematopoietic stem and progenitor cell specification

Authors: Tim Petzold, Sarah, Brivio, Tanja, Linnerz, Masakatsu Watanabe, Holger Gerhardt, Julien Y. Bertrand

I am happy to tell you that your manuscript has been accepted for publication in Biology Open, pending our standard publication integrity checks. It was accepted on 18 June 2025.